# Scalable electrosynthesis of commodity chemicals from biomass by suppressing non-Faradaic transformations

Hua Zhou[1,2,3,7], Yue Ren[1,7], Bingxin Yao[1], Zhenhua Li [1], Ming Xu[1], Lina Ma[4], Xianggui Kong[1], Lirong Zheng[5], Mingfei Shao [1,3] & Haohong Duan [2,6] ✉

Electrooxidation of biomass platforms provides a sustainable route to produce valuable oxygenates, but the practical implementation is hampered by the severe carbon loss stemming from inherent instability of substrates and/or intermediates in alkaline electrolyte, especially under high concentration. Herein, based on the understanding of non-Faradaic degradation, we develop a single-pass continuous flow reactor (SPCFR) system with high ratio of electrode-area/electrolyte-volume, short duration time of substrates in the reactor, and separate feeding of substrate and alkaline solution, thus largely suppressing non-Faradaic degradation. By constructing a nine-stacked-modules SPCFR system, we achieve electrooxidation of glucose-to-formate and 5-hydroxymethylfurfural (HMF)-to-2,5-furandicarboxylic acid (FDCA) with high single-pass conversion efficiency (SPCE; 81.8% and 95.8%, respectively) and high selectivity (formate: 76.5%, FDCA: 96.9%) at high concentrations (formate: 562.8 mM, FDCA: 556.9 mM). Furthermore, we demonstrate continuous and kilogram-scale electrosynthesis of potassium diformate (0.7 kg) from wood and soybean oil, and FDCA (1.17 kg) from HMF. This work highlights the importance of understanding and suppressing non-Faradaic degradation, providing opportunities for scalable biomass upgrading using electrochemical technology.

Biomass represents an abundant and sustainable carbon reservoir to replace nonrenewable fossil resource for chemicals and fuels production, matching the target of decarbonization and defossilization in industrial sector[1–4]. Unlike fossil feedstocks, biomass typically feature high oxygen content (40−45 wt.%)[5], making it an ideal starting material to produce valuable oxygenates, such as formic acid[6], 2,5-furandicarboxylic acid (FDCA)[7–9], furoic acid[10], and saccharic acids[11]. However, owning to the inherently high reactivity of oxygen-containing functional groups in biomass derivatives (e.g., aldehyde,

ketone), the upgrading technologies often suffer from undesired degradation under extreme reaction conditions (e.g., high-temperature, basic or acidic medium), which is further aggravated for concentrated feedstock[12,13]. A long-recognized example is sugar degradation to a portfolio of organic acids in alkaline solution, owning to the rearrangement of polyhydroxy aldehyde/ketone structure of sugar under base catalysis[14,15], a first-order reaction with respect to the sugar concentration[16]. Another academically and industrially important reaction is 5-hydroxymethylfurfural (HMF) oxidation to FDCA,

[1]State Key Laboratory of Chemical Resource Engineering, College of Chemistry, Beijing University of Chemical Technology, Beijing 100029, China. [2]Department of Chemistry, Tsinghua University, Beijing 100084, China. [3]Quzhou Institute for Innovation in Resource Chemical Engineering, Quzhou 324000, China. [4]Shandong Institute of Petroleum and Chemical Technology, Dongying 257061, China. [5]Institute of High Energy Physics, Chinese Academy of Sciences, Beijing 100049, China. [6]Haihe Laboratory of Sustainable Chemical Transformations, Tianjin 300192, China. [7]These authors contributed equally: Hua Zhou, Yue Ren. ✉e-mail: hhduan@mail.tsinghua.edu.cn

which is used for the production of bio-plastics[17], as exemplified by the Avantium process[18]. This reaction is also suffering from HMF degradation to humins via decomposition and self-polymerization[12], especially for concentrated HMF feedstock that is favorable for dimerization and further oligomerization[19].

To lessen the above issue, most academic research were performed at low-concentration (e.g., HMF concentration of 0.5–2.1 wt.%)[12] to obtain high selectivity and yield of targeted product. However, industrially-relevant operation often requires high-concentration products for cost-effective and less energy-intensive product separation and chemicals recovery[20]. Recently, stabilization of the reactive functional groups in biomass derivatives (e.g., sugars[19,21], HMF[12], lignin[22]) was demonstrated as an effective strategy to suppress degradation. However, extra procedures to introduce and remove the protecting agent complicate the process and increase the operating costs, calling for simple and efficient strategy to realize concentrated biomass valorization.

In this respect, electrooxidation is emerging as an advantageous tool for upgrading biomass platforms derived from raw biomass into chemicals and fuels (Fig. 1a), because of mild reaction conditions (room temperature and ambient pressure), no external oxidants, and ideally driven by renewable electricity[11,23–25]. Meanwhile, green hydrogen can be obtained at cathode via hydrogen evolution reaction, which is regarded as an important energy carrier of intermittent renewable energy[26–29]. Tremendous efforts were made in converting biomass-derivatives (e.g., glycerol, glucose, HMF and lignin) into valuable chemicals[9,10,28,30–33], and significant achievements were achieved on catalyst design to enhance activity and selectivity[34,35]. Taking HMF

oxidation to FDCA as an example, near quantitative conversion and good selectivity (>90%) have been achieved (Supplementary Table 1). However, a comprehensive survey of these literature reveals that most studies were performed in dilute HMF solution (≤10 mM) and small-volume (2–30 mL). These reaction conditions are prevalent in reported electrocatalytic transformations of biomass platforms (Supplementary Tables 1, 2), limiting the process scalability (Fig. 1b). The underlying reason for not using high-concentration and large-volume biomass platforms was less discussed in previous literature, which presumably can be attributed to the aforementioned degradation issue. Moreover, biomass electrooxidations were often performed in strong alkaline electrolyte (typically 1 M KOH) for enhancing current density and activity[10,32,36,37], but typical biomass derivatives (e.g., glucose and HMF) are vulnerable to degradation in alkaline medium (Supplementary Fig. 1). Therefore, it remains largely unexplored to tackle the degradation issue (denoted as non-Faradaic degradation), which represents a major obstacle for electrolysis of biomass platforms under industrially-relevant conditions (such as high-concentration and large-volume).

Recently, a few studies devoted to addressing this issue in biomass electrooxidation[38–40]. Krebs and co-workers[39] demonstrated a stabilization strategy by converting labile HMF to alkali-stable 5-hydroxymethyl-2-furancarboxylic acid (HMFCA) and dihydroxymethylfuran (DHMF) via base-catalyzed Cannizzaro reaction, both of which can be further oxidized to FDCA via electrolysis. However, the overall carbon balance is still unsatisfactory (~80%) when using 250 mM of feedstock. In addition, Latsuzbaia and colleagues[40] achieved the reported highest FDCA concentration (5 wt.%, ~350 mM,

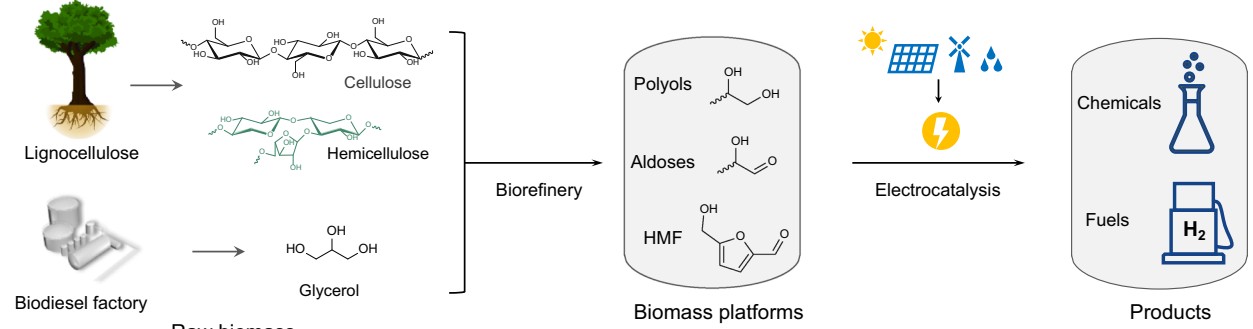

**a** Electrocatalytic biomass upgrading

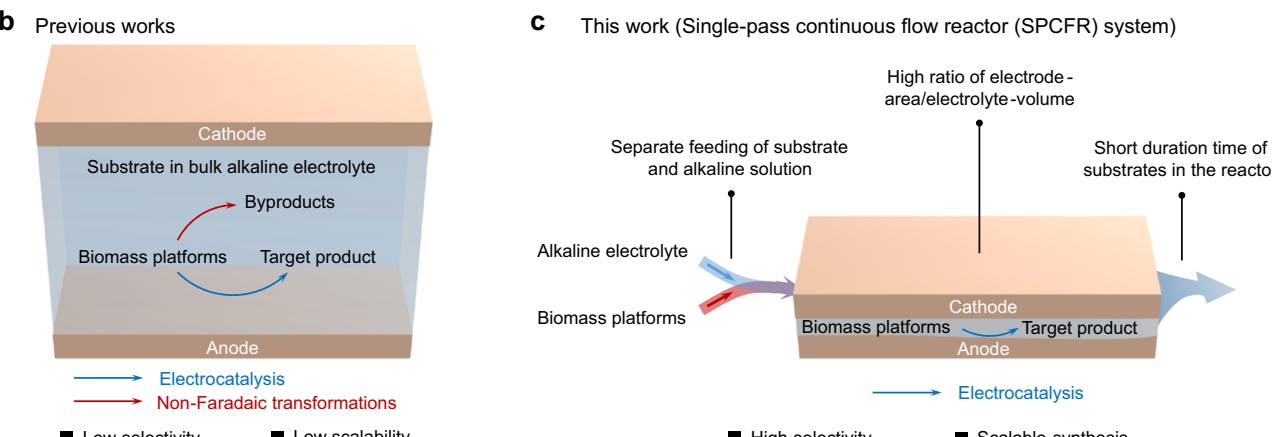

**b** Previous works

**c** This work (Single-pass continuous flow reactor (SPCFR) system)

**Fig. 1 | Electrocatalytic upgrading of biomass platforms. a** Electrocatalytic biomass platforms upgrading toward chemicals and fuels by integrating with upstream biorefinery processes. **b** Scheme of conventional electrooxidation of biomass platforms in a batch reactor, in which undesired non-Faradaic degradation often take place in bulk alkaline electrolyte, resulting in low selectivity towards target product and low scalability. **c** The SPCFR system developed in this work, showing three key features that suppress non-Faradaic degradation, enabling high selectivity towards target product and scalable synthesis at high concentrations.

~70% yield) using 10 wt.% HMF as the starting material in a neutral electrolyte to suppress degradation, a high productivity was maintained in a circulated flow reactor by continuous addition of NaOH. Despite of these advances, it remains challenge to achieve facile and selective electrooxidation of concentrated biomass platforms.

Herein, we reported a facile system engineering strategy to solve the carbon loss issue (degradation) when performing alkaline electrolysis of typical biomass platforms (e.g., glycerol, glucose and HMF), enabling large-scale production of value-added oxygenates with high concentration. First, we demonstrated that the carbon loss is originated from the unstable properties of substrate and/or reaction intermediates in alkaline electrolyte via non-Faradaic side-reactions, representing the main obstacle for process scalability. On this basis, we developed a single-pass continuous flow reactor (SPCFR) system to suppress non-Faradaic degradation, showing high ratio of electrode-area/electrolyte-volume, short duration time of substrates in the reactor, and separate feeding of substrate and alkaline solution (Fig. 1c). We further constructed a stacked SPCFR system (consisting of nine stacked modules) with geometric electrode area up to 270 cm², achieving electrooxidation of glucose-to-formate and HMF-to-FDCA with high single-pass conversion efficiency (SPCE; 81.8%, 95.8%, respectively) and high selectivity (formate: 76.5%, FDCA: 96.9%) at high concentrations (formate: 562.8 mM, FDCA: 556.9 mM). As proof-of-concept, we demonstrated kilogram-scale synthesis of potassium diformate (KDF, a commercial animal feed additive; 0.7 kg) from wood and soybean oil, and FDCA (1.17 kg) from HMF. Moreover, by eliminating the interference of non-faradaic products, we unveil that aldoses are the real intermediates in electrooxidation of glucose-to-formate, which contrasts the most adopted view of aldonic acids as the intermediates. This work provides opportunities for scaling up of electrochemical biomass conversion technology and insights for reaction pathway understanding.

## Results

### Non-Faradaic degradation in scaling-up biomass platforms electrooxidation

Given the intrinsic unstable properties of typical biomass-derivatives (Supplementary Fig. 1), we hypothesized that the non-Faradaic degradation could be a major obstacle for the practical implementation of biomass electrooxidation technology, especially in alkaline electrolyte[36,39]. Therefore, we initiated the study by evaluating the trade-off between targeted electrocatalytic conversion and unwanted degradation during electrolysis of biomass platforms in alkali electrolyte in conventional batch reactor. Cobalt oxyhydroxide (CoOOH) was selected as the anodic catalyst because it is widely used in alkaline oxygen evolution reaction (OER)[41,42] and biomass derivatives electrooxidations[43,44]. By using an electrodeposition method[32], a γ-phase CoOOH with array morphology was fabricated on a nickel foam matrix (CoOOH/NF), which was characterized with complementary techniques (Supplementary Figs. 2–4). Then, electrochemical measurements were conducted in 1 M KOH for electrooxidation of various biomass-derived polyhydroxy compounds (Supplementary Figs. 5, 6), including aldoses (glucose, xylose) and polyols (sorbitol, xylitol, erythritol, glycerol, ethylene glycol (EG)) over CoOOH/NF. Linear sweep voltammetry tests reveal that CoOOH/NF exhibits electro-activity toward oxidation of these substrates prior to OER (Supplementary Fig. 7), giving formate as the main product as demonstrated by ¹H nuclear magnetic resonance spectra (¹H NMR, Supplementary Fig. 8). As a complement to extensively studied glycerol or glucose electrooxidation[28,44], these results highlight the generality of electrochemistry for oxidative C − C bond cleavage of molecules containing vicinal hydroxyl or aldehyde groups to carboxylates (e.g., formate), a class of important reactions have broad applications in upgrading waste biomass and plastics[32,45-47], as well as organic synthesis[48].

To demonstrate how non-Faradaic degradation affects process scale-up, glycerol (a byproduct in biodiesel production) and glucose (monomer from cellulose saccharification) were selected for further study, because they represent typical biomass-derived polyol and aldose, respectively. The electrolysis was performed in an undivided batch reactor (Supplementary Fig. 6) using CoOOH/NF (1 cm² of geometric area) as anode, with different substrate concentrations and electrolyte volumes. The products were quantified by high-performance liquid chromatography (HPLC, Supplementary Fig. 9). As shown in Fig. 2a, the catalytic performances are strongly dependent on both the substrate concentration and electrolyte volume. Specifically, in glycerol electrooxidation, a good formate selectivity (>97%) was obtained in scenario A: low-concentration glycerol (10 mM) in large-volume (50 mL), and in scenario C: high-concentration glycerol (100 mM) in small-volume (5 mL), both are popular configurations used in literatures[11,49,50]. In evident contrast, formate selectivity decreased to 81.9% under larger scale (scenario B: high-concentration glycerol (100 mM) in large-volume (50 mL)). The decrease of selectivity when scaling-up glycerol electrooxidation is due to the non-Faradaic degradation of glycerol-derived intermediates (i.e., glyceraldehyde and glycolaldehyde) in bulk electrolyte via base-catalyzed dehydration and Cannizzaro rearrangement (i.e., non-Faradaic side-reactions) to lactate[26,36,37], as evidenced by the increased lactate fraction (9.0%) in the products (Fig. 2a). This observation is more obvious in glucose electrooxidation, in which formate selectivity was only 30.4% under scenario B, with the generation of a portfolio of organic acids (e.g., lactate, glycerate, glycolate, gluconate). The formate selectivity under scenario B is much lower than that obtained under scenario A or C (>84%; Fig. 2b). The severe non-Faradaic degradation of glucose can be ascribed to its polyhydroxy aldehyde structure that is vulnerable to degradation in alkaline medium[15,51]. Preliminary kinetic analysis reveals that approximate 54% of the converted glucose fraction was consumed by non-Faradaic side-reactions during electrolysis (in scenario B at 1.5 V vs RHE), indicating that the unwanted non-Faradaic degradation even surpasses the targeted electrocatalytic conversion (see details in Supplementary Note 1, Supplementary Figs. 11–13). The similar phenomenon was also observed in electrooxidation of other polyhydroxy compounds (sorbitol, xylitol, xylose, erythritol, EG) (Supplementary Fig. 14, Table 4). These results suggest that non-Faradaic degradation of feedstock and/or intermediates is a non-negligible issue that limiting process scaling-up of electrooxidation of polyhydroxy compounds to formate.

The above observations motivated us to examine another extensively studied biomass upgrading reaction, HMF oxidation to FDCA (Fig. 2c), a model reaction for oxidative dehydrogenation of alcohols[7]. Similarly, FDCA selectivity dramatically decreased from >87.7% to 37.8% when using high-concentration HMF (200 mM) in large-volume (50 mL) electrolyte (scenario B, Fig. 2c, Supplementary Fig. 15). Dark-brown humins was observed and isolated (inset in Fig. 2c, Supplementary Fig. 16), suggesting the occurrence of self-polymerization as the dominant non-Faradaic side-reaction[12,19]. To demonstrate its general applicability, we further evaluated the HMF oxidation performances with nickel foam (NF) supported nickel phosphide (Ni₂P/NF, Supplementary Fig. 17), which was previously demonstrated as a selective catalyst for HMF electrooxidation to FDCA[52]. A similar catalytic trend was observed when replacing CoOOH/NF with Ni₂P/NF (Supplementary Fig. 18), indicating that the carbon loss issue in HMF electrooxidation is not mainly related to the selection of catalyst. These results unambiguously show that HMF electrooxidation to FDCA at industrial-relevant requirements (high-concentration and large-volume) is also hampered by non-Faradaic degradation, representing a great challenge for industrialization of biomass electrooxidation technologies.

A simplified scheme to illustrate the reactions taking place in electrooxidation of biomass substrates/intermediates is shown in Fig. 2d,

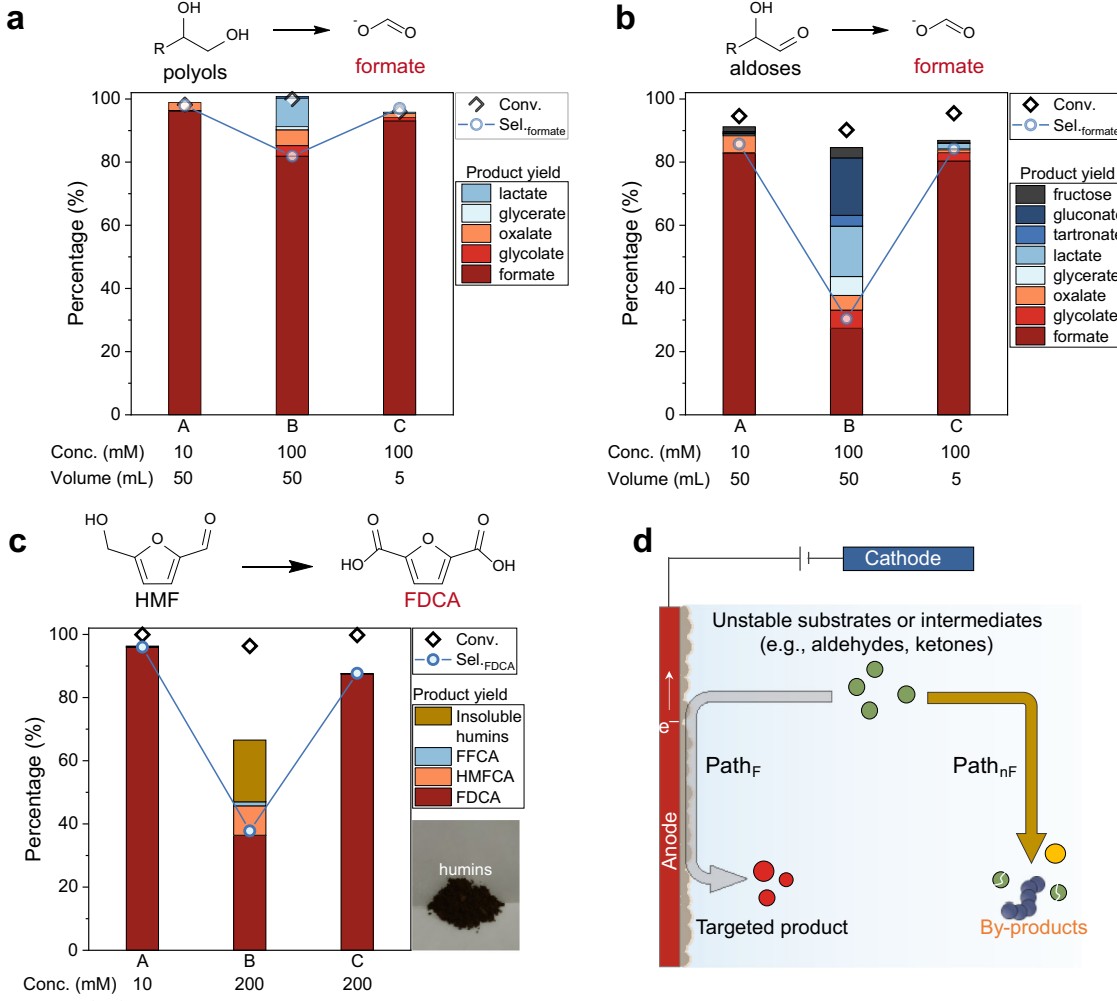

**Fig. 2 | Trends of typical biomass platforms electrooxidation in conventional batch processing.** Catalytic performances in electrooxidation of glycerol (**a**), glucose (**b**), and HMF (**c**) as function of feedstock concentration and electrolyte volume at 1.5 V vs RHE using CoOOH/NF as the anode. **d** Scheme of competition between electrocatalytic reaction over anode surface and non-Faradaic transformations in bulk electrolyte.

where two competitive and independent paths usually existed, including targeted electrocatalytic one (Path$_F$) over anode surface and undesired non-Faradaic one (Path$_{nF}$) in bulk electrolyte. The overall catalytic performances rely on the kinetic balance between these two paths, where Path$_F$ can be facilitated by efficient catalyst and cell configurations, and Path$_{nF}$ is dependent on multiple factors: (1) inherent properties of biomass-derived feedstock and intermediates; (2) process parameters (cell configurations, substrate concentration, electrolyte pH and temperature, duration time of reactions), which we anticipate can be regulated by system engineering.

**Design of SPCFR system to suppress non-Faradaic degradation**
With the understandings of non-Faradaic degradation, we expect that system engineering with delicate cell configuration may facilitate Path$_F$ and suppress non-Faradaic degradation in scaling-up for biomass platforms electrooxidation. To promote Path$_F$, it is critical to improve the ratio of electrode-area to electrolyte-volume (A/V), according to Eq. (1) for electrochemical reactor (refs. [53,54]):

$$X = 1 - \exp\frac{-k_m At}{V} \qquad (1)$$

Where X is the fractional reactant conversion via electrolysis, $k_m$ is the mass transport coefficient, t is the reaction time. The exponential

relationship between A/V and X suggests that a high conversion efficiency can be reached in a short duration time in an electrolyzer with high A/V value. This relationship between cell configurations and performances accounts for the high selectivity in electrooxidation of biomass feedstocks under scenario C (Fig. 2a–c), wherein small-volume electrolyte was used (thus scenario C shows a higher A/V value than scenario B under the same concentration), thus Path$_F$ was promoted. In principle, a high A/V value can be readily achieved in conventional flow reactors (Fig. 3a) for biomass electrooxidation[55–57].

Nevertheless, even electrolyzer with high A/V value was used, biomass feedstocks and electrolyte were often mixed together and circulated through the anode chamber in this reaction system (Fig. 3a)[55–57], without considering the instability of substrates and intermediates in electrolyte that results in severe side-reactions, remaining as an unmet challenge for scalable electrolysis. Similar challenge was also encountered in synthetic organic chemistry, which can be tackled by flow chemistry strategy, featuring separate storage of reactive substrate and solvent, and rapid transformation of reactive substrate into targeted product in a reactor with high surface-to-volume ratio[58–60].

Leveraging these successful design principles, we constructed a SPCFR system (Fig. 3b) with following features: (1) a high A/V value (2.5 cm$^2$ electrode per mL electrolyte) to facilitate the reaction stream of Path$_F$. (2) Separate supply of substrates and alkali electrolyte into

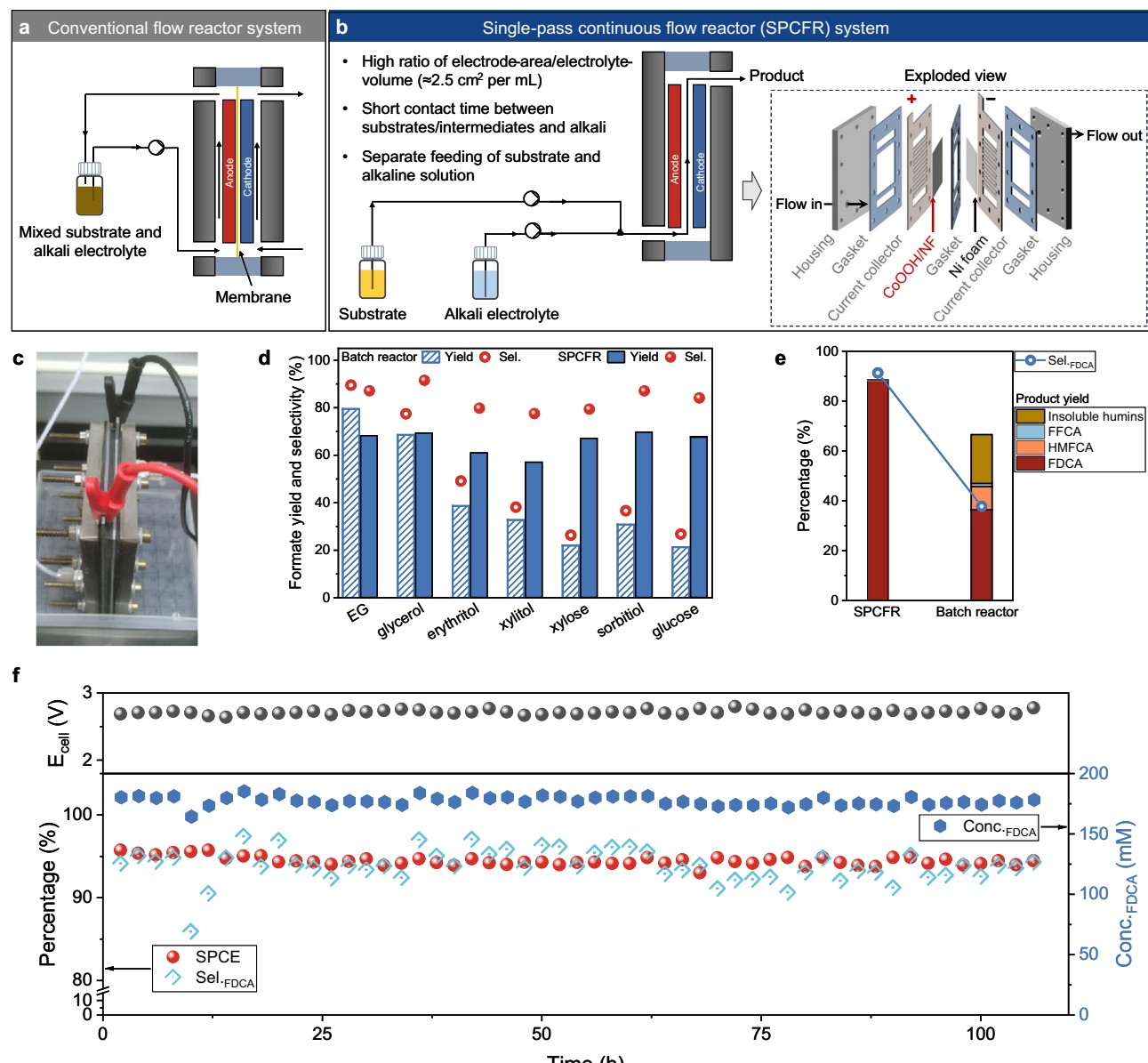

**Fig. 3 | Steering non-Faradaic side-reactions by SPCFR system. a** Traditional flow reactor system. **b** Our designed SPCFR system. **c** Photograph of single-module SPCFR. Catalytic performances of SPCFR system and batch reactor for electrooxidation of polyhydroxy compounds-to-formate (**d**) and HMF-to-FDCA (**e**). **f** Continuous synthesis of FDCA from 200 mM HMF at 5 A with a flow rate of 1.06 mL min⁻¹. Configurations of the single-module SPCFR: 30 cm² geometric area of CoOOH/NF and nickel foam were used as anode and cathode, respectively. The electrolyte volume in the reactor was maintained to be about 12 mL. Ionic exchange membrane is absence in the reactor, considering the negligible effect between anodic biomass electrooxidation and cathodic HER[9,11].

the reactor. (3) A short duration time of electrolyte in the reactor. (4) A high SPCE at the reactor effluent. Features (2)–(4) can be rationalized as to decrease the contact time between the alkali-vulnerable substrates/intermediates and alkaline medium. With this system engineering, we envisage to facilitate Path_F and suppress Path_nF, thus achieving high product selectivity under high-concentration in alkaline electrolyte.

To validate our design, a single-module SPCFR system (Fig. 3c and Supplementary Fig. 19) was constructed and employed for the electrooxidation of glucose (100 mM) in alkaline condition (1 M KOH). Notably, this system achieves high SPCE (80.2%) with high formate selectivity (83.8%) and Faradaic efficiency (FE) (89.6%) at flow rate of 1.98 mL min⁻¹ and current of 3 A (Supplementary Figs. 20, 21), outpacing that in batch reactor (formate selectivity of 30.4%) and previous literature (formate selectivity <55%)[61,62]. The effectiveness of the SPCFR system was then corroborated by selective conversion of

other biomass-derived polyhydroxy compounds into formate, which cannot be accomplished in a batch reactor (Fig. 3d, Supplementary Figs. 22, 23). The high formate selectivity in the SPCFR system is owing to the suppression of non-Faradaic transformations to other organic acids, as evidenced by HPLC chromatograms (Supplementary Fig. 24). We believe the high selectivity of formate is beneficial for the downstream product purification and separation in practical application.

Furthermore, we evaluated the SPCFR system in electrooxidation of high-concentration HMF (200 mM). The results show high SPCE (96.6%) with high FDCA selectivity (91.3%) at a flow rate of 0.79 mL min⁻¹ and current of 3 A. The total selectivity of other detectable products is 0.70%, including DFF (0.13%), HMFCA (0.38%), and FFCA (0.19%). We speculate that the carbon loss (that is 7.7%, according to carbon balance of 92.3%) is mainly attributed to the formation of humins, with fewer generation compared with the results in

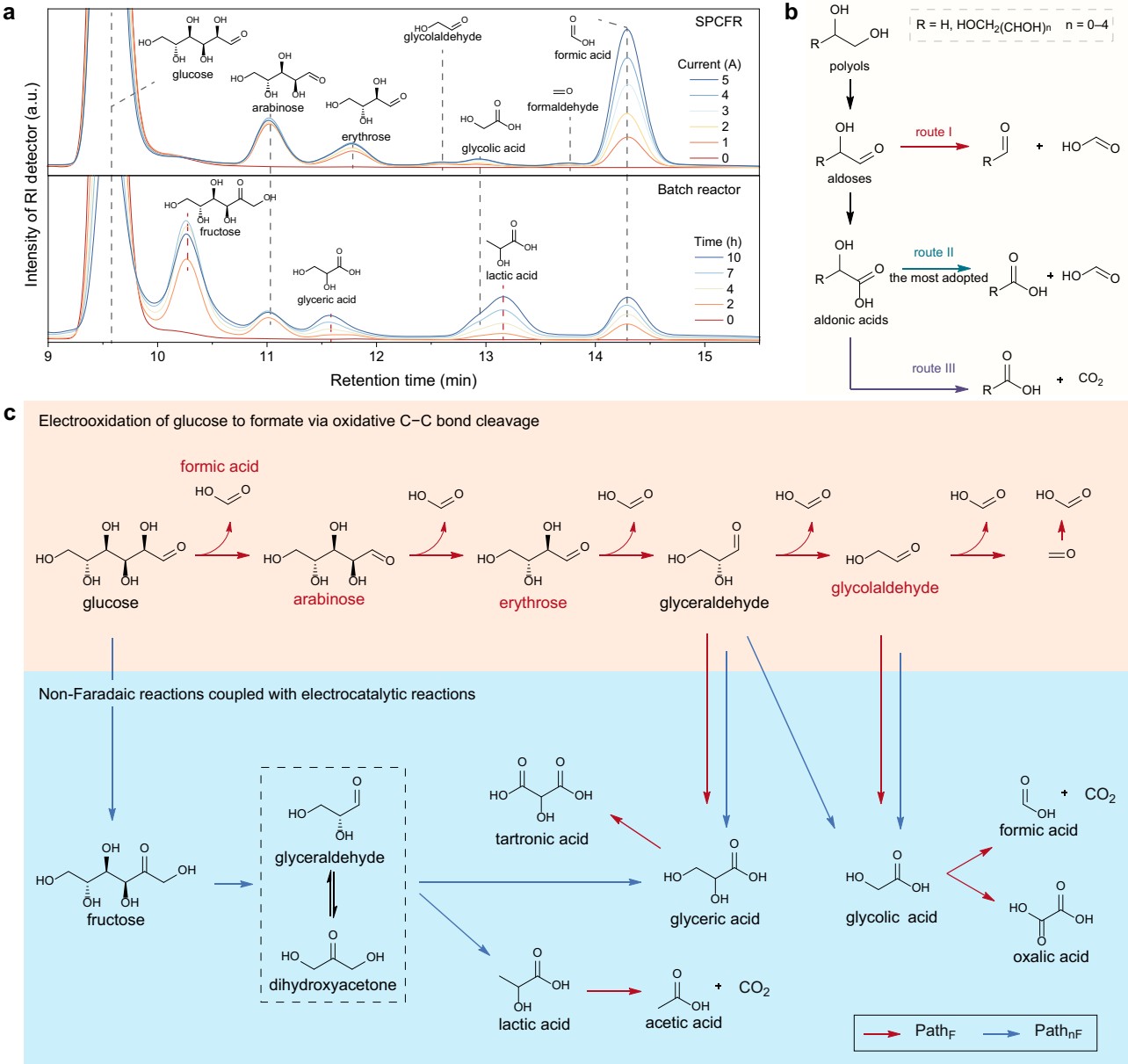

**Fig. 4 | Mechanistic investigation. a** HPLC chromatograms of the electrolyte of SPCFR and batch reactor. **b** Reported reaction pathways for C − C bond cleavage of polyhydroxy compounds. **c** Proposed reaction maps for alkaline glucose electrolysis. Red arrows indicate electrocatalytic reactions, and blue arrows indicate non-Faradaic reactions. The compounds labeling in red were detected by HPLC.

batch reactor, which is challenging to quantify (Fig. 3e, Supplementary Figs. 26, 27). As a result, high-purity (>99%) FDCA was produced and isolated in a continuous manner that is suitable for industrially-relevant operation (Supplementary Figs. 28–30). No obvious decay of catalytic performances and catalyst structure was observed for more than 100-h at current of 5 A (Fig. 3f, Supplementary Figs. 30, 31). Specifically, a stable SPCE (~94%), cell voltage (~2.7 V), and high FDCA selectivity (~95%) were maintained in the continuous operation (Fig. 3f). The robustness of the SPCFR system was also demonstrated in continuous glucose electrooxidation to formate (Supplementary Fig. 32).

**Probing the key intermediates**
Probing the key intermediates during biomass electrooxidation is critical to in-depth understand how the SPCFR system achieves substantially improved selectivity compared to the conventional batch reactor. As it is extremely difficult to detect highly reactive aldehydes

in alkaline electrolyte, we further ameliorated the system by integrating SPCFR with following immediate acid quenching and HPLC analysis, aiming to detect short-lifetime species (Supplementary Note 2 and Fig. 33). We carried out glucose-to-formate as a model reaction at a high flow rate (11.4 mL min⁻¹) and different currents (0–7 A), to reduce the degradation of unstable species in the effluent of SPCFR.

By employing this ameliorated system, a series of aldehydes (including aldoses (arabinose, erythrose, and glycolaldehyde) and formaldehyde) and glycolate (in the form of glycolic acid) were observed in HPLC results (Fig. 4a (top), Supplementary Fig. 34a). In obvious contrast, if immediate acid quenching was not performed, the above aldehydes and unreacted glucose would degrade rapidly or even totally consumed (i.e., glycolaldehyde, formaldehyde) in minutes, which is accompanied with the increase of fructose, lactate, glycolate, and glycerate (Supplementary Fig. 35). Similar results were observed in the HPLC chromatograms using a batch reactor, in which most of the aforementioned aldehydes (except

arabinose) were missing, and fructose, lactate, glycolate, and glycerate were observed (Fig. 4a (bottom), Supplementary Fig. 34b). This can be explained by the rapid non-Faradaic degradation of aldehyde intermediates in conventional batch reactor. To demonstrate aldehydes were also in situ generated in conventional batch reactor but were consumed by non-Faradaic reactions, on-line differential electrochemical mass spectrometry (DEMS) was conducted in batch-type glucose electrooxidation (see details in Supplementary Fig. 36), and formaldehyde was indeed detected. Collectively, these results suggest that aldehydes are the real intermediates in glucose electrooxidation to formate. The aldehydes preferentially degrade into acids in bulk electrolysis in batch reactor, which can be largely suppressed by employing the SPCFR system.

By surveying the literature, the reported reaction pathways for oxidative C−C bond cleavage of polyhydroxy compounds to produce formate are summarized in Fig. 4b, including the aldose route (route I[28,63]) and the most adopted aldonic acid route (route II[34,35,49,64,65]). According to above experimental results, we speculate that the non-Faradaic degradation products (i.e., aldonic acids) was possibly misinterpreted as the intermediate for formate production in a myriad of previous publication. By combining above HPLC results and $^{13}C1$-labeling experiments (see discussion in Supplementary Note 3), we can exclude route II for glucose-to-formate, and rationalize that the reaction was initiated from C1−C2 bond position of glucose (Fig. 4a (up)). This is well agreement with our previous theoretical calculation that glucose electrooxidation is initiated at C1−C2 position to give arabinose and formate, owing to the smallest bond order among the five C−C bonds in glucose molecule[44] (see details in Supplementary Note 3).

Based on these findings, we proposed electrooxidation of glucose to formate follows a processive α-scission pathway via aldehyde as the key intermediates, rather than aldonic acids (red arrow flow in Fig. 4c). In contrast, in SPCFR system without immediate acid quenching or in batch reactor, non-Faradaic degradation would be coupled into the transformations, with the generation of other organic acids (Fig. 4c (bottom)). The oxidative C−C bond cleavage of aldonic acids generate $CO_2$ (route III in Fig. 4b). The demonstration of reaction pathway that follows aldose route, rather than the most adopted aldonic acid route, can be extended to other polyhydroxy compounds electrooxidation (e.g., glycerol-to-formate; see discussion in Supplementary Fig. 40, Note 3).

As exemplified by glucose electrolysis, the coexistence of non-Faradaic reactions in parallel with electrocatalytic pathway not only limit the process scalability, but also interference the understanding of real reaction pathway. Our preliminary mechanistic studies demonstrate that the SPCFR system prevents the short-lifetime intermediates from non-Faradaic degradation, providing opportunities to scaling-up and resolving a long-debating views of aldose route or most-adopted aldonic acid route for C−C bond cleavage of polyhydroxy compounds to produce formate. We believe a more optimized SPCFR system integrated with advanced in-line/on-line analytical tools can be developed in the future, which can be employed as an automated and sensitive instrument for detection of reactive intermediates in electrocatalytic transformation of organic compounds (Supplementary Fig. 41).

## Large-scale electrooxidation using a stacked SPCFR system

For practical implementation of biomass electrooxidation technologies, industrially-relevant metrics—such as high productivity, high product concentration, and compatibility with crude feedstocks—should be considered. To improve productivity, we amplified the reactor from single module to nine stacked modules, establishing a stacked SPCFR system with geometric electrode area up to 270 cm² (Fig. 5a, Supplementary Figs. 42, 44). By preliminarily optimizing current, flow rate, and temperature of feedstock solution, we achieved high productivity of formate in electrooxidation of glucose, delivering high SPCE (81.8%), good formate selectivity (76.5%), and high FE (91.7%) at current of 15 A (Supplementary Table 5). As a result, formate solution (562.8 mM) with space-time-yield (STY) of 256.6 mmol h⁻¹ (corresponding to 11.8 g h⁻¹) coupling with $H_2$ production (>99.9% purity) with STY of 279.8 mmol h⁻¹ (corresponding to 0.56 g h⁻¹) were continuously produced using this stacked SPCFR system (Supplementary Figs. 43, 44), achieving co-production of biomass-derived valuable chemicals and $H_2$ fuel. Furthermore, the reaction system can be linearly scaled up by tandem design without diminishing catalytic performances (see discussion in Supplementary Figs. 45, 46), showing great potential of assembling modules in parallel or in cascade for pilot-scale application (Supplementary Fig. 47). Note that the electrolyte temperature increased under high current owing to Joule heating effect (e.g., ~70 °C at 15 A), thus non-Faradaic degradation was accelerated by two-orders of magnitude (specifically, 163 times; Supplementary Fig. 48), accounting for the low formate selectivity (<65%) for reactions without temperature management (entries 4−6 of Supplementary Table 5). Therefore, further studies on temperature management are highly demanded, by means of reducing ohmic resistance or integrating cooling system.

To demonstrate the compatibility of SPCFR to real biomass derivatives, we prepared low-grade polyhydroxy compounds including carbohydrates (produced from biorefinery processes of lignocellulose) and crude glycerol (produced from biodiesel production[23,66]) as the feedstocks for electrooxidation in the stacked SPCFR system (Fig. 5b). Raw sugars were obtained from birch wood via biomass pretreatment and enzymatic saccharification (see details in Supplementary Figs. 49−51, Note 4) and then fed into the stacked SPCFR system. Eventually, a high formate selectivity (73.1%) at 81.5% SPCE of sugars was afforded (Supplementary Fig. 51). Crude glycerol was obtained by KOH-catalyzed soybean oil transesterification (Supplementary Fig. 52, Note 4). By employing the SPCFR system, a high formate selectivity (76.3%) at 83.1% SPCE of glycerol was obtained. These results demonstrate the high compatibility of SPCFR to real biomass feedstocks. Furthermore, considering the high cost of formic acid separation from alkaline electrolyte[67], the as-obtained formate was acidified by formic acid to obtain potassium diformate (KDF). KDF is a value-added commodity (Supplementary Table 6) for inhibiting the propagation of harmful bacteria such as *E. coli, Salmonella* spp., etc[67,68], which has been authorized as a safe alternative of antibiotic for promoting animal growth[69]. Finally, columnar KDF crystals was obtained at near kilogram-scale (0.7 kg, Fig. 5c, Supplementary Fig. 53), and identified by XRD (PDF#22-0813), NMR, and HPLC analysis (Supplementary Figs. 53−56).

Producing high-concentration product is another important metric for scalability of biomass electrooxidation technology. For example, in electrooxidation of HMF to FDCA, it remains challenging to reach FDCA concentration >500 mM (Supplementary Table 1). This motivated us to test high-concentration (300−800 mM) HMF electrooxidation using the stacked SPCFR system. To our delight, starting with 600 mM HMF, a high concentration (556.9 mM) of FDCA was produced with good selectivity (96.9%), presenting a STY of 76.2 mmol h⁻¹ (corresponding to FDCA productivity of 11.9 g h⁻¹). Even higher concentration of FDCA (up to 602.7 mM) was obtained, although with decreased selectivity (80.7%, Supplementary Fig. 57a). Under the optimized HMF concentration (600 mM), high-concentration FDCA (530−560 mM) was continuously produced at high SPCE (>95%) for over 50 h using the stacked SPCFR system (Fig. 5d), outperforming state-of-the-art reports on HMF electrooxidation (Fig. 5e; 144 publications were summarized). Finally, kilogram-scale (1.17 kg) FDCA was isolated in our laboratory (Fig. 5f). These results demonstrated the potential of the stacked SPCFR system for industrially-relevant applications.

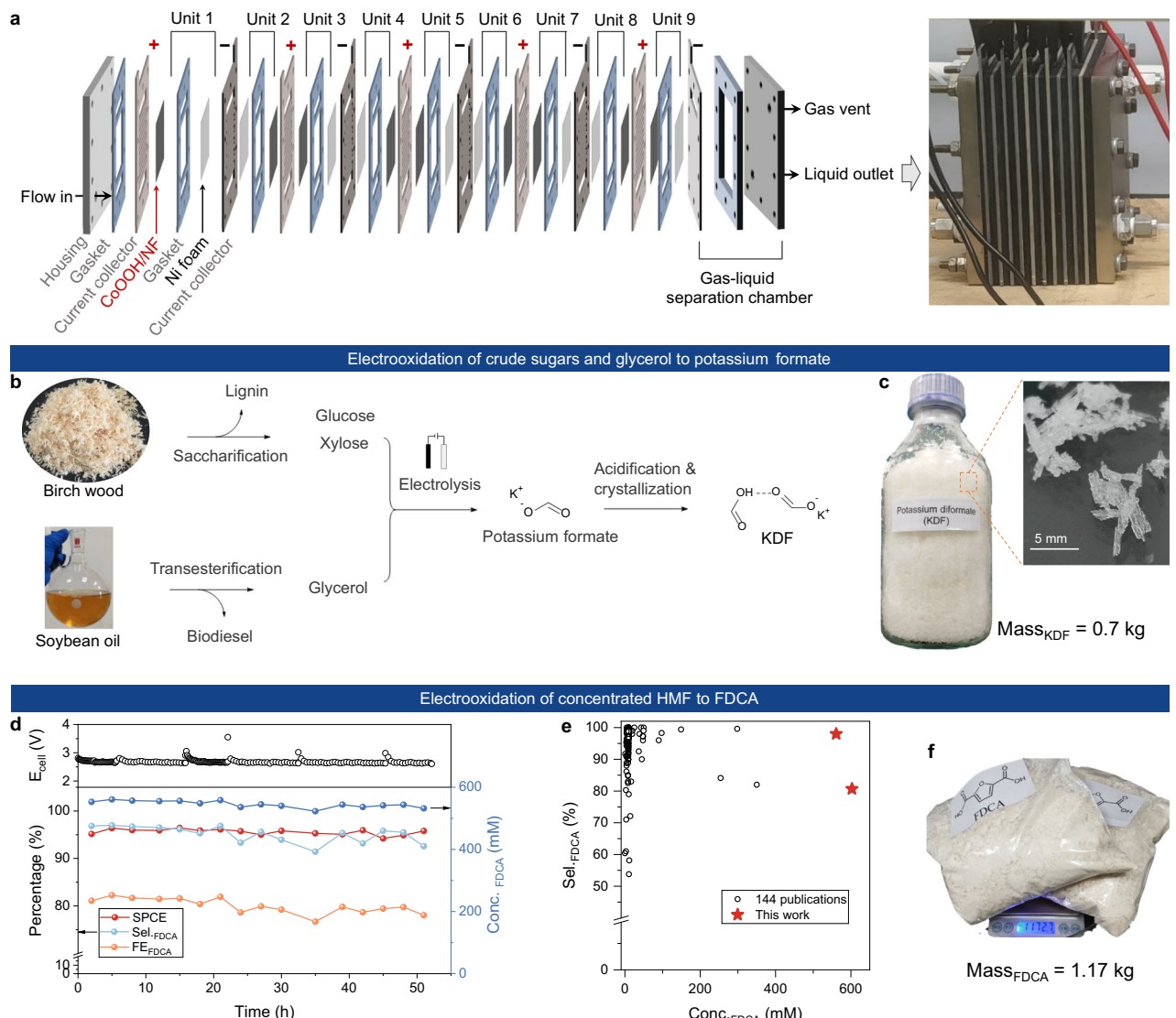

**Fig. 5 | Scalable electrosynthesis by using stacked SPCFR system. a** Illustration and photograph of stacked SPCFR consisting of nine stacked modules. **b** Scheme of KDF synthesis from raw biomass-derived polyhydroxy compounds. **c** Photographs of isolated KDF (0.7 kg), showing columnar crystals. **d** Continuous electrooxidation of concentrated HMF (600 mM) at 15 A and a flow rate of

2.28 mL min⁻¹. The experiments were performed in the daytime for safety considerations; thus, the fluctuation of cell voltage was observed when restart the system. **e** Comparison of FDCA concentration and selectivity of this work and state-of-the-art works. **f** Photograph of the isolated FDCA (1.17 kg) from the experiments in Fig. 3f and Fig. 5d.

## Discussion

In this work, we disclosed that non-Faradaic degradation of feedstock and/or intermediates is a general and non-negligible challenge in alkaline electrolysis of typical biomass platforms, limiting scaling-up of process and misleading reaction pathway understanding. An efficient SPCFR system was developed to improve the selectivity in electrooxidation of biomass-derived polyhydroxy compounds to formate, and HMF to FDCA, especially for high concentration in large volume. By suppressing non-faradaic degradation, the SPCFR system provides evidence to support aldose route for oxidative C – C bond cleavage of polyhydroxy compounds, ruling out the most adopted aldonic acid route. We further constructed a stacked SPCFR system (nine-stacked-modules with geometric electrode area of 270 cm²) to effectively bridge upstream biorefinery processes and downstream product separation for large-scale synthesis of valuable commodity chemicals (i.e., KDF, FDCA). This work highlights the opportunities of system

engineering to realize electrocatalytic biomass valorization coupling with H₂ production in a scalable manner.

## Methods

### Chemicals and materials

NF (1.6 mm thick) was purchased from MTI. KOH (99.99%, metal basis), Co(NO₃)₂.6H₂O (99.99%, metal basis), D-fructose (99%), and D-[1-¹³C] glucose (99%) were purchased from Aladdin. D-glucose (99%), D-sorbitol (98%), D-xylose (99%), D-xylitol (98%), erythritol (99%), glycerol (99%), ethylene glycol (99%), glycolic acid (99%), formic acid (98%), methanol (anhydrous, 99.99%) and activated charcoal (Norit® SA2) were purchased form Innochem. D-[U-¹³C6] glucose (99%) was purchased from Cambridge. D-[1-¹³C] gluconic acid (99%, sodium salt) was purchased from Omicron. Soybean oil (reagent grade) was purchased from Meryer. Cellic® CTec2 (Novozymes), glyceric acid (99%, calcium salt), and lactic acid (85%) were purchased from Sigma-Aldrich. HMF (99%) was provided by Zhejiang

Sugar Energy Technology Co., Ltd. All chemicals were used as received unless otherwise stated.

## Characterization

XRD patterns were recorded on a Bruker D8 Advance diffractometer. SEM images were recorded on Zeiss Supra 55. TEM images were acquired on FEI TECNAI G[2]. Raman spectra were LabRAM Aramis instrument. X-ray photoelectron spectroscopy (XPS) was performed on Kratos Axis Supra instrument. [1]H NMR spectra were recorded on a Bruker 400 M NMR instrument. HPLC was performed with an Agilent 1260 equipped with UV and RI detector.

## Synthesis of CoOOH

The CoOOH nanosheet array was growth and supported on NF according to the previous method[67]. Specifically, α-phase Co(OH)$_2$ was electrodeposited on NF in 100 mM Co(NO$_3$)$_2$ electrolyte by applying a constant current of −80 mA for 300 s. After water rinsing, the as-obtained Co(OH)$_2$/NF was activated in 1 M KOH to synthesis γ-phase CoOOH/NF[32,67].

## Electrochemical evaluations

All electrochemical measurements for anodic oxidation were performed in an undivided cell on an electrochemical workstation (CHI 760E or CHI 1130 C, CH Instruments, Inc.). The CV, LSV, and pulse voltammetry tests were carried out in a three-electrodes system, using saturated Ag/AgCl and platinum foil as reference and counter electrodes, respectively. All potentials measured against Ag/AgCl were converted to the reversible hydrogen electrode (RHE) scale using $E_{RHE} = E_{Ag/AgCl} + 0.197 + 0.059$ pH without iR compensation.

## Electrosynthesis of formate

For the electrooxidation of glucose and other polyhydroxy compounds in batch reactor, three-electrode set-up was applied, using CoOOH/NF (1 cm$^2$), saturated Ag/AgCl, and platinum foil as working, reference, and counter electrodes, respectively. The reaction was carried out in a beaker with 50 mL 1 M KOH and 100 mM substrates at 1.5 V vs RHE for specific time.

For the transformation of glucose and other polyhydroxy compounds in flow cell, two-electrode set-up was applied, using CoOOH/NF (30 cm$^2$) and NF (30 cm$^2$) as anode and cathode, respectively (details are illustrated in Supplementary Fig. 19). The aqueous solution of polyhydroxy compounds (200 mM for single modular flow cell, 300 mM for cell stack) and KOH (2 M) were separated and mixed before pumping into the cell. Then, the flow cell is worked at constant current mode driven by a direct-current power supply (A-BF®).

## Isotope labeling experiments

To reduce the usage of [13]C labeled substrates, the isotope labeling experiments were carried out in a vial, containing 4 mL 1 M KOH with 100 mM substrate. Two-electrode set-up was applied, using CoOOH/NF and Pt wire as anode and cathode, respectively, at cell voltage of 1.5 V. The electrolyte was taken at different charge transfer and immediately neutralized by 0.5 M H$_2$SO$_4$ solution, then analyzed by [1]H NMR.

## Fractionation of lignocellulosic biomass.

A formic acid (FA) promoted biomass fractionation protocol was applied in this work as illustrated in Supplementary Fig. 49. Specifically, 20 g dried birch wood was added into a 1 L flask with 400 mL aqueous FA solution (FA: water = 7:3, v/v) and the mixture was refluxed for 3 h with stirring at 1500 rpm. During this process, the hemicellulose component was hydrolyzed into xylose and lignin was dissolved into the solution. When the slurry was cooled to room temperature, the cellulose

pulp was separated by filtration and sequentially washed with DI water, dilute KOH (0.1 M), and DI water. The filtrate was evaporated under reduced pressure to recovery FA solution. Then, the residual solid was washed with water to dissolve sugars and a small fraction of lignin oils and to obtain solid lignin fraction. The aqueous stream was further extracted ethyl acetate to yield lignin oil and sugar solution.

The cellulose pulp was further saccharified by a commercial enzyme (Cellic® CTec2) to yield glucose. In detail, cellulose pulp was dispersed in sodium acetate buffer (50 mM, pH 5.0) with 10 wt.% of cellulose, then cellulase (5 mg protein/g cellulose) was added and the reaction processed in an incubator (50 °C, 200 rpm).

The content of cellulose, hemicellulose, and lignin in native lignocellulose and cellulose pulp were analysed according to the protocol developed by National Renewable Energy Laboratory (NREL)[70].

## Biodiesel and crude glycerol production.

To integrate with downstream electrolysis, we adopted a KOH catalysis method for biodiesel production[71–73], as illustrated in Supplementary Fig. 52a. Specifically, 400 g soybean oil was added into a 1 L flask and preheated to 65 °C on a heating plate. Then a freshly prepared methanol solution (134 mL) with 6.5 g KOH were added into the preheated oil. The mixture was kept at 65 °C and stirred at 1500 rpm for 2 h to ensure the completion transesterification reaction. After that the reaction mixture was allowed to cool down and equilibrate in a separatory funnel. The lower phase consisted of glycerol, KOH, methanol was collected for further electrocatalytic transformation. While the upper methyl esters phase was refined by acid neutralization, washing, distillation, and dehydration processes to obtain biodiesel (345.2 ± 15.3 g), corresponding to 86.3 wt.% of weight yield.[73] The amount of glycerol in crude glycerol was determined to be (39.8 ± 0.06 g) by HPLC analysis, corresponding to 9.79 wt.% (g$_{glycerol}$/g$_{triglyceride}$). Finally, the process was repeated for four times.

## Synthesis of KDF

The obtained electrolyte (-8 L) from the electrooxidation of crude sugars and glycerol was first acidified with FA to pH 3 and decolorized with activated charcoal (2 g). Then the solution was concentrated to -1 L by rotary evaporation under reduced pressure at 60 °C, which was mixed with 1 L ethanol and crystallized at −20 °C for two days. Finally, the KDF crystal was obtained by filtration, washing with anhydrous ethanol and dried in vacuum oven.

## Calculations

The FE was calculated with the following equation:

$$FE(\%) = 100\% \times \frac{\text{mole of product}}{\text{total charge passed}/(n \times 96485 \text{ C mol}^{-1})} \quad (2)$$

Where 96,485 C mol$^{-1}$ is the faraday constant. $n$ is the average number of electron transfer for the formation of per product, $n = 2$ for converting glucose and xylose, $n = 7/3$ for sorbitol, $n = 12/5$ for xylitol, $n = 10/3$ for erythritol, $n = 8/3$ for glycerol, $n = 3$ for EG, $n = 4$ for methanol, which calculated according to the oxidation numbers of carbon atom in these molecules (Supplementary Table 7). $n = 6$ for converting HMF to FDCA.

The yield of products in this work refers to carbon yield, which was calculated with the following equation:

$$Yield(\%) = 100\% \times \frac{\text{mole of product} \times \text{carbon numbers in product}}{\text{mole of substrate} \times \text{carbon numbers in substrate}}$$

$$(3)$$

The product selectivity was calculated with the following equation:

$$Selectivity(\%) = 100\% \times \frac{product\ yield}{substrate\ conversion} \qquad (4)$$

## Data availability

The data for figures in this study are provided in the Source Data file. The data that support the findings of this study are available in the paper and its Supplementary Information or from the corresponding author upon request. Source data are provided with this paper.

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

## Acknowledgements

H.D. acknowledges the support from Beijing Natural Science Foundation (Grant No. JQ22003), the National Natural Science Foundation of China (NSFC, Grant Nos. 22325805, 21978147, 21935001) and the Haihe Laboratory of Sustainable Chemical Transformations. H.Z. acknowledges the support from Fundamental Research Funds for the Central Universities (No. buctrc202211), National Key R&D Program of China (Grant No. 2022YFA1504200), and NSFC (Grant Nos. 22308015, 22288102). Z.L. acknowledges the support from NSFC (Grant No. 21935001) and National Key R&D Program of China (Grant No. 2022YFB4002700). L.M. acknowledges the support from NSFC (No. 22105015). The authors thank the BL1W1B in the Beijing Synchrotron Radiation Facility (BSRF).

## Author contributions

H.D. supervised this project and revised the manuscript. H.Z. conceived the concept and wrote the manuscript. Y.R. and B.-X.Y. performed the experiments. M.X. and L.Z. analyzed XAFS data. Z.L., L.M., X.K. and M.S. discussed and reviewed the manuscript.

## Competing interests

The authors declare no competing interests.
