## [Peer Review File · Nature Communications]

Reviewer #1 (Remarks to the Author):

In this manuscript, the authors investigated the limitations in process scale-up of biomass platforms electrooxidation and proposed a general and efficient system engineering strategy for the preparative electrosynthesis of valuable products from typical biomass platform chemicals. They show that degradation of instable substrate and reaction intermediates in alkaline electrolyte, especially at high concentration, is one of the major challenges for scaling-up of reaction system of electrocatalytic biomass upgrading. On this foundation, they constructed a single-pass continuous flow reactor system for selective and scalable upgrading of biomass derivatives, such as glucose, HMF. The electrocatalytic conversion of organic substrates (e.g., biomass derivatives) is an active research area nowadays. Unlike most research focusing on efficient catalyst synthesis, the authors demonstrated the importance and opportunities of system engineering for process scalability, and unveiled the reaction pathways that is often misunderstood by non-Faradaic reaction interference. The manuscript is well organized and well supported by a suite of experimental data. I believe this work will have a notable impact for the development of electrocatalytic biomass upgrading towards practical application. Therefore, I recommend its publication in Nature Communications after the authors address the following minor issues:

1. CoOOH was used as the model catalyst to investigate non-Faradaic degradation issue and subsequent system engineering. I wonder if similar trends in Fig. 2 can be observed for other extensively used anodic catalysts, such as nickel-based materials?
2. Page 7, the authors need explain the sentence "Preliminary kinetic analysis reveal that approximate 54% of glucose was consumed via non-Faradaic reactions during electrolysis (i.e., scenario B at 1.5 V vs RHE), ...".
3. It is interesting to detect unstable reaction intermediates with the assistance of the developed SPCFR system. However, the description of experimental process in the manuscript is too simple. The authors should describe more details about the experimental procedure for reproducibility.
4. The authors should check the order of figures. For example, In Supplementary Note 4, Supplementary Fig. 41 is not in consistent with the context. Should it be Supplementary Fig. 42?
5. Some sentences need to be more concise, such as "a γ -phase CoOOH with array morphology CoOOH was fabricated on nickel foam (CoOOH/NF),..." on Page 6, the second "CoOOH" can be deleted.
6. The legend of Supplementary Fig. 41b is unclear. The authors should clearly demonstrate the mean of each curve in the Figures.
7. The abbreviations are not uniform throughout the manuscript and SI. e.g., "BR" in Supplementary Fig. 18, "CFR" in Supplementary Fig. 25.
8. Some recent publications on electrooxidation of polyhydroxy compounds should be added into Supplementary Table 2, such as, Adv Mater 2023, e2300935; Appl Catal B: Environ 2022, 305, 121082; Adv Mater, 2023, 35(4): 2370022; Nat Commun, 2022, 13, 5848; Nat Commun, 2022, 13, 3777.

Reviewer #2 (Remarks to the Author):

In this manuscript, the authors developed a simple and efficient single-pass continuous flow reactor (SPCFR) system to suppress non-faradaic degradation of substrate and improve the selectivity in electrooxidation of biomass, such as polyhydroxy compounds to formate, and HMF to FDCA. A series of experimental evidence solidly support the mechanistic insight of aldose route for oxidative C–C bond cleavage of polyhydroxy compounds, rather than the most adopted aldonic acid route. Moreover, the kilogram-scale electrosynthesis of KDF and FDCA are fascinating, showing the potential scalable electrosynthesis. The manuscript could be considered to publish on Nature Communications after addressing the following problem.

1. In this SPCFR system, a good high single-pass conversion efficiency (SPCE) was achieved. If liquid stream goes through the SPCFR system one more time via peristaltic pump, could the conversion efficiency further improve?
2. In Fig. 3b and line 253, what is the basis for absenting the ionic exchange membrane in the designed SPCFR system? Whether the reaction intermediate and target product of anodic oxidation will be reduced or hydrogenated at the cathode?
3. In Fig. 3d and line 271, is it possible for other by-products to be formed since the FDCA selectivity is 92.3% not closes to 100%?
4. In supplementary Figure 17, the conversion and formate selectivity increase as the flow rate decreases, which means that lower flow rate is beneficial to more formation of formate. However, the FE of formate shows the opposite trend at the same condition, why? The similar phenomenon also displays in supplementary Figure 21, why?
5. It is suggested that the authors provide detailed characterizations of CoOOH/NF anode catalyst after stability testing of Fig. 3e.
6. In supplementary Figure 31 and 32, the integration of NMR spectra is unclear, it is better to provide clearer one.
7. Some writing mistake can be found in Fig. 3b, such as “the mose adopted”. Please thoroughly check the full text carefully.
8. What is the solubility of HMF like the high concentration up to 800 mM in the aqueous phase? Is a cosolvent needed?
9. If the liquid stream flow rate ratio (i.e. the ratio of substrate to KOH) change via two peristaltic pumps with different flow rates in this SPCFR system, what will be happened for the performance? They were

fixed to be same via one peristaltic pump like in supplementary Figure 36. Is this parameter studied in more detail?

10. How does the gas-liquid separation chamber work in the stacked SPCFR system in supplementary Figure 36? What is the amount and purity of the hydrogen that generated and separated?

Rev #3

I am very positive about the publication of this article. The data presented here is relevant to the field and the conclusions are well-supported by the experiments. Below are some comments.

- 1) During the oxidation of the biomass-derived molecules, green hydrogen is produced in the cathode. It called my attention that the authors have not mentioned the importance of the production of this energy vector.
- 2) In this paper, the separation and quantification of the products are extremely relevant. The main drawback of this paper is that the authors did not clearly explain the HPLC protocol used to run the samples. They should also show which products are they able to indeed separate.
- 3) About the mechanism, you could mention something more than simply “providing experimental evidence for our previous theoretical calculation”. I think that the calculations you published are extremely valuable, but the level of detail of them is much deeper than what these results can show. Here you show that you break the C-C bond and the importance and reactivity of the aldehydes, which is really valuable, but you are not giving much evidence for mechanistic details. To link these observations (I mean, the products generated in different conditions) to the detailed mechanism is too much. What indeed can give stronger insights into the mechanism are, for example, Raman in situ experiments like that you performed in the same paper and/or a simpler experiment like that shown in figure 17 of this manuscript:
<https://chemrxiv.org/engage/chemrxiv/article-details/63f64e9032cd591f12534067>
The importance/reactivity of the aldehydes have been shown in several articles but most of them use PGM electrodes. I am not suggesting including the references but maybe they help improving the discussion.

<https://www.sciencedirect.com/science/article/pii/S0013468618325581?via%3Dihub>

- 4) Authors achieved the following “By preliminary optimization of current and flow rate, and by

temperature management of feeding solution, we achieved high productivity of formate in electrooxidation of glucose, delivering high SPCE (81.8%), good formate selectivity (76.5%), and high FE (91.7%) at current of 15 A (Supplementary Table 4).” Considering 100% of faradaic efficiency, I think informing the green hydrogen productivity is also important.

5) In the Large-scale electrosynthesis experiments, the authors state that the cell working temperature is around 70°C and that they must work on temperature management. I think that instead of cooling down the system, it is maybe better to optimize the method at this relatively high temperature.

6) Again, about the production of value-added molecules. The authors informed the concentration of products. As they know the flow in the cell, they could inform the mass of products obtained per unit of time, for instance, per hour.

7) To close, I would like to suggest the authors stop using Pt CE. The Pt atoms can migrate to the working electrode and deposit there.

Response to Reviewer #1:

In this manuscript, the authors investigated the limitations in process scale-up of biomass platforms electrooxidation and proposed a general and efficient system engineering strategy for the preparative electrosynthesis of valuable products from typical biomass platform chemicals. They show that degradation of instable substrate and reaction intermediates in alkaline electrolyte, especially at high concentration, is one of the major challenges for scaling-up of reaction system of electrocatalytic biomass upgrading. On this foundation, they constructed a single-pass continuous flow reactor system for selective and scalable upgrading of biomass derivatives, such as glucose, HMF. The electrocatalytic conversion of organic substrates (e.g., biomass derivatives) is an active research area nowadays. Unlike most research focusing on efficient catalyst synthesis, the authors demonstrated the importance and opportunities of system engineering for process scalability, and unveiled the reaction pathways that is often misunderstood by non-Faradaic reaction interference. The manuscript is well organized and well supported by a suite of experimental data. I believe this work will have a notable impact for the development of electrocatalytic biomass upgrading towards practical application. Therefore, I recommend its publication in Nature Communications after the authors address the following minor issues:

Comment 1: CoOOH was used as the model catalyst to investigate non-Faradaic degradation issue and subsequent system engineering. I wonder if similar trends in Fig. 2 can be observed for other extensively used anodic catalysts, such as nickel-based materials?

Response: We thank the reviewer for this insightful comment. According to the

suggestion, the manuscript has been revised based on the following additional experiments. A nickel foam supported phosphide ($\text{Ni}_2\text{P}/\text{NF}$, revised Supplementary Fig. 17) was selected for catalytic test, because it was previously demonstrated as a selective catalyst for HMF electrooxidation to FDCA (*Angew. Chem. Int. Ed.* 2016, 55, 9913–9917). As shown in revised Supplementary Fig. 18, FDCA selectivity dramatically decreased when HMF electrolyte with higher concentration (200 mM) and larger volume (50 mL) was used. Specifically, FDCA selectivity decreased from 92.9% (scenario A) and 86.7% (scenario C) to 46.6% (scenario B). Therefore, we can conclude that the similar trend shown in Fig. 2 was observed when well-studied anodic catalyst was used (revised Supplementary Fig. 18), indicating that the carbon loss issue in HMF electrooxidation is not mainly related to the selection of catalyst.

Based on above discussion, we revised the Manuscript and Supplementary Information as follows:

“To demonstrate its general applicability, we further evaluated the HMF oxidation performances with nickel foam supported nickel phosphide ($\text{Ni}_2\text{P}/\text{NF}$, Supplementary Fig. 17), which was previously demonstrated as a selective catalyst for HMF electrooxidation to FDCA⁵². A similar catalytic trend was observed when replacing CoOOH/NF with $\text{Ni}_2\text{P}/\text{NF}$ (Supplementary Fig. 18), indicating that the carbon loss issue in HMF electrooxidation is not mainly related to the selection of catalyst.”

(Please see Page 7 in the revised Manuscript)

Revised Supplementary Figure 17. Characterization of Ni_2P . **a** SEM images of Ni_2P nano-array. Inset: enlarged region showing interconnected nano-particles. **b** High-resolution TEM image of Ni_2P nano-particle. **c** XRD pattern of $\text{Ni}_2\text{P}/\text{NF}$.

As shown in Supplementary Figure 17a, SEM images revealed nano-array structure of Ni₂P, and the nano-array is composing of interconnected nano-particles (inset). TEM combined with XRD characterizations (Supplementary Fig. 17b, c) further confirmed the Ni₂P structure.

Revised Supplementary Figure 18. Catalytic performances of HMF electrooxidation in different scenarios (different feedstock concentrations and electrolyte volumes) at 1.5 V vs RHE using Ni₂P/NF as the anode.

As shown in Supplementary Fig. 18, FDCA selectivity dramatically decreased when HMF electrolyte with higher concentration (200 mM) and larger volume (50 mL) was used. Specifically, FDCA selectivity decreased from 92.9% (scenario A) and 86.7% (scenario C) to 46.6% (scenario B). Therefore, we can conclude that the similar trend in Fig. 2 was observed when extensively used anodic catalyst was used (Supplementary Fig. 18), indicating that the carbon loss issue in HMF electrooxidation is not mainly related to the selection of catalyst.

Comment 2: Page 7, the authors need explain the sentence “Preliminary kinetic analysis reveal that approximate 54% of glucose was consumed via non-Faradaic reactions during electrolysis (i.e., scenario B at 1.5 V vs RITE), ...”.

Response: This is a critical comment. Indeed, the original expression was ambiguous. In our experiments, we evaluated reaction velocity of glucose conversion in 50 mL 1 M KOH solution with 100 mM glucose under electrochemical conditions (at 1.5 V vs

RHE, corresponding to scenario B at 1.5 V vs RHE in the manuscript) or under chemical conditions (without applying bias nor electrocatalyst, to study non-Faradaic reactions).

The results show that, under electrochemical conditions, the apparent reaction velocity of glucose conversion was estimated to be $5.44 \times 10^{-6} \text{ mol L}^{-1} \text{ s}^{-1}$ (Supplementary Fig. 11a), which is contributed by both electrocatalytic and non-Faradaic side-reactions. In contrast, under chemical conditions, the glucose reaction velocity was measured to be $1.64 \times 10^{-6} \text{ mol L}^{-1} \text{ s}^{-1}$, which is contributed by non-Faradaic side-reactions, namely base-catalyzed glucose degradation (Supplementary Fig. 12). Based on the above catalytic results, we estimate that about 54% of the converted glucose fraction was contributed by non-Faradaic side-reactions during electrolysis (in scenario B in the manuscript).

To clearly present these results, we revised the Manuscript and Supplementary Information as follows:

1. We rewrite the sentence in the revised Manuscript:

“Preliminary kinetic analysis reveals that approximate 54% of the converted glucose fraction was consumed by non-Faradaic side-reactions during electrolysis (in scenario B at 1.5 V vs RHE), indicating that the unwanted non-Faradaic degradation even surpasses the targeted electrocatalytic conversion (see details in Supplementary Note 1, Supplementary Figs. 11–13).” (Please see Page 7 in the revised Manuscript)

2. We revised Supplementary Note 1 in the Supplementary Information:

“Furthermore, the reaction velocity of non-Faradaic reaction (v_{hF}), that is base-catalyzed glucose degradation, was calculated to be $1.64 \times 10^{-6} \text{ mol L}^{-1} \text{ s}^{-1}$ (Supplementary Fig. 12; in 1 M KOH solution without applying bias nor electrocatalyst). Based on the catalytic results under electrochemical conditions (at 1.5 V vs RHE) and under chemical conditions (without applying bias nor electrocatalyst), we estimate that about 54% of the converted glucose fraction was contributed by non-Faradaic side-reactions during electrolysis (scenario B at 1.5 V vs RHE).” (Please see Page 3 in the Supplementary Information)

Comment 3: It is interesting to detect unstable reaction intermediates with the

assistance of the developed SPCFR system. However, the description of experimental process in the manuscript is too simple. The authors should describe more details about the experimental procedure for reproducibility.

Response: We appreciate the reviewer for this valuable suggestion. We have presented the details of experimental configurations in the figure captions of revised Supplementary Figure 33, and reorganized the information in figure to clearly show the experiment process and the key points. (*Please see Page 30 in the Supplementary Information*).

The revised Supplementary Figure 33 is shown below:

Revised Supplementary Figure 33. Scheme of probing unstable intermediates during electrocatalytic glucose oxidation.

The system is divided into three parts (from left to right):

i Separate storage: The feedstock solutions of electrolyte (2 M KOH) and glucose (200 mM) were separately stored to suppress base-catalyzed non-Faradaic degradation to generate organic acids.

ii Rapid transformation: A single-module SPCFR (Supplementary Fig. 19) was employed for electrocatalytic GOR to formate using a mixed electrolyte composing of 1 M KOH and 100 mM glucose at a flow rate of 11.4 mL min⁻¹. The SPCFR system enables rapid transformation of glucose, affording detectable reaction intermediates for

subsequent HPLC analysis. The SPCFR system also shortens the duration time of glucose and intermediates in the reactor, hence diminishing non-Faradaic degradation. In addition, the SPCFR can be operated at different currents (0–7 A) to detect the variation of possible reaction intermediates.

iii Fast neutralization and analysis: The electrolyte at the outlet of SPCFR was collected and immediately quenched (that is, neutralized) by a dilute acid (0.5 M H₂SO₄) to stabilize the reactive intermediates. The products were then immediately analyzed by HPLC. By doing this way, the reactive intermediates (such as aldehyde intermediates) can be stabilized without being transformed into organic acids by base-catalyzed degradation.

Comment 4: The authors should check the order of figures. For example, In Supplementary Note 4, Supplementary Fig. 41 is not in consistent with the context. Should it be Supplementary Fig. 42?

Response: We thank the reviewer for this meticulous comment. We carefully checked and corrected the order of all the figures and tables.

Comment 5: Some sentences need to be more concise, such as “a γ -phase CoOOH with array morphology CoOOH was fabricated on nickel foam (CoOOH/NF),...” on Page 6, the second “CoOOH” can be deleted.

Response: We thank the reviewer for the helpful suggestion. We checked the manuscript and rewrote the following sentences:

“By using an electrodeposition method³², a γ -phase CoOOH with array morphology was fabricated on a nickel foam matrix (CoOOH/NF), which was characterized with complementary techniques (Supplementary Figs. 2-4).” (*Please* see Page 6 in the revised Manuscript)

“a Electro-catalytic biomass platforms upgrading toward chemicals and fuels by integrating with upstream biorefinery processes.” (*Please* see Page 4 in the revised Manuscript)

Comment 6: The legend of Supplementary Fig. 41b is unclear. The authors should clearly demonstrate the mean of each curve in the Figures.

Response: This is an important comment. We replotted the following figures:

Revised Supplementary Figure 50b Yield of glucose via enzyme-catalyzed cellulose hydrolysis as a function of reaction time. Error bars correspond to the standard deviation of three measurements.

Revised Supplementary Figure 16b HMF conversion and product formation as a function of reaction time at 1.5 V vs RHE. HMF conversion (red rectangular points and line). Product yields: circular points and lines. Carbon balance: black triangular points and line. FDCA selectivity: blue triangular points and line. FDCA faradaic efficiency: circular points and line.

Comment 7: The abbreviations are not uniform throughout the manuscript and

SI. e.g., “BR” in Supplementary Fig. 18, “CFR” in Supplementary Fig. 25.

Response: We appreciate the meticulous comment. We checked the manuscript and Supplementary Information, and unified the abbreviations as “batch reactor” and “SPCFR” throughout the figures and text. For example, Supplementary Figs. 18 and 25 have been revised as below:

Revised Supplementary Figure 22. Conversion of biomass-derived sugars and polyols (with increased carbon numbers) and FE to formate using batch reactor and SPCFR. (Please see Page 23 in the Supplementary Information)

Revised Supplementary Figure 30. Set-up for continuous HMF oxidation in SPCFR. **a** Schematic illustration of the electrocatalytic HMF oxidation in SPCFR. **b** Photograph of set-up. (*Please see Page 28 in the Supplementary Information*)

Comment 8: Some recent publications on electrooxidation of polyhydroxy compounds should be added into Supplementary Table 2, such as, *Adv Mater* 2023, e2300935; *Appl Catal B: Environ* 2022, 305, 121082; *Adv Mater*, 2023, 35(4): 2370022; *Nat Commun*, 2022, 13, 5848; *Nat Commun*, 2022, 13, 3777.

Response: We thank the reviewer for providing the valuable reference. 10 additional papers on electrooxidation of organic substrates to formate (including the abovementioned references) have been added in entries 20–29 of the revised Supplementary Table 2.

Revised Supplementary Table 2. Electrocatalytic oxidation of biomass-derived polyhydroxy compounds and methanol to formate

Entry	Substrate	Catalyst	Reactor	Electrolyte a			FE _{formate} (%)	Conv. (%)	Sel. _{formate} (%)	Conc. _{formate} (mM)	Ref.
				Base	Substrate	Volume					
				Conc. (M)	Conc. (mM)	(mL)					
1	glucose			1	150		91.7	81.8	76.5	562.8	
2	glucose			1	100		91.5	80.4	84.0	405.2	
3	sorbitol			1	100		98.8	80.0	86.3	414.2	
4	xylose			1	120		85.1	84.5	79.2	401.5	
5	xylitol	CoOOH	Single-pass flow cell	1	120		89.3	73.5	77.4	341.3	This
6	erythritol			1	150		93.2	76.4	79.6	364.9	work
7	glycerol			1	200		82.5	75.7	91.4	415.1	
8	ethylene glycol			1	300		85.6	78.3	87.0	408.3	
9	xylose	NiOOH	Circulated	0.2	10	NG	NG	85	48	20.4	
10	glucose	RuO ₂ /Ti	Flow cell	0.5	10	NG	NG	65	38	≈14.82	183
11	glucose	Cu	H-cell	0.1	40	NG	NG	NG	54.2	NG	30
12	glucose	NiFe-1	H-cell	1	100	5	86	NG	87	522	184

13	glycerol	HEA-CoNiCuMnMo	Circulated Flow cell	1	100	NG	92	NG	NG	NG	19
14	glycerol	Ni-Mo-N/CFC	Undivided cell	1	100	5	95	100	93	279	27
15	glycerol	NiCo hydroxide	H-cell	1	100	50	≈100	≈90	94.3	254.6	185
16	glycerol	CoMoO ₄	H-cell	1	100	10	92.7	47.3	67.6	95.9	16
17	glycerol	CuCo-oxide	Undivided cell	0.1	100	NG	NG	≈36	≈85	≈91.8	15
18	glycerol	NC/Ni-Mo-N	Undivided cell	1	100	≈15	NG	≈85	≈95	≈242.25	186
19	glycerol	CuCo ₂ O ₄	Undivided cell	0.1	100	2	89.1	79.7	80.6	192.71	14
20	glycerol	NiVRu-LDHs NAs/NF	Circulated Flow cell	1	100	NG	80	NG	NG	NG	187
21	glycerol	Ni(OH) ₂	H-cell	0.1	25	14	84	NG	NG	≈15.77	188
22	glycerol	NiCo	H-cell	1	100	50	NG	≈89	94.3	≈84.03	189
23	glycerol	CuCo-oxide	Undivided cell	0.1	100	NG	NG	≈35.6	≈85.5	≈30.43	190
24	glycerol	Ni ₃ N/Co ₃ N-NWs	Circulated Flow cell	1	100	NG	≈96.4	NG	NG	NG	191
25	glycerol	Bi-Co ₃ O ₄	H-cell	1	100	40	97.05	NG	97.01	NG	192
26	glycerol	ZnFe ₂ O ₄	H-cell	1	500	20	NG	8.5	74.69	31.59	13

27	ethylene glycol	CoNi _{0.25} P/NF	H-cell	1	300	40	82.5	100	90.2	270.6	10
28	methanol	Ni ₃ S ₂ /CNTs	Undivided cell	1	1000	500	>95	NG	NG	NG	193
29	methanol	NiMn-LDH	H-cell	1	3000	NG	96.8	NG	NG	NG	194

Response to Reviewer #2:

In this manuscript, the authors developed a simple and efficient single-pass continuous flow reactor (SPCFR) system to suppress non-faradaic degradation of substrate and improve the selectivity in electrooxidation of biomass, such as polyhydroxy compounds to formate, and HMF to FDCA. A series of experimental evidence solidly support the mechanistic insight of aldose route for oxidative C-C bond cleavage of polyhydroxy compounds, rather than the most adopted aldonic acid route. Moreover, the kilogram-scale electrosynthesis of KDF and FDCA are fascinating, showing the potential scalable electrosynthesis. The manuscript could be considered to publish on Nature Communications after addressing the following problem.

Comment 1: In this SPCFR system, a good high single-pass conversion efficiency (SPCE) was achieved. If liquid stream goes through the SPCFR system one more time via peristaltic pump, could the conversion efficiency further improve?

Response: We appreciate for this valuable suggestion.

1. Based on this comment, we constructed a two-tandem-modules reaction system for glucose electrooxidation, in which the reaction solution was feed into the first reactor module, and the liquid stream was then feed into the second one (revised Supplementary Fig. 45a).

Under an optimized conditions (i.e., current of 3 A, flow rate of 1.98 mL min⁻¹ obtained from Supplementary Fig. 20), good catalytic results (conversion of 80.2%, formate selectivity of 83.8% and FE of 89.6%) were obtained after electrolyte passed through the first module (revised Supplementary Fig. 45b). After the liquid stream passed through the second module, higher conversion efficiency (94%) was achieved, but formate selectivity (27.8%) and FE (16.8%) decreased. This can be explained by the overoxidation of formate to carbonate, as evidenced by the formation of large number of CO₂ when the electrolyte was acidified (pretreatment for HPLC analysis).

In terms of catalytic performance, electrolyte that passes through multiple modules at a given flow rate is equivalent to that passes through one reactor at a

lower flow rate. This is because when multiple modules are adopted, the duration time of electrolyte in the reactor proportionally increases, with same result obtained in one reactor with lower flow rate.

Based on this understanding, we anticipate that the reaction in one reactor may deliver similar catalytic trend (higher conversion but lower formate selectivity and FE) if the flow rate is lower than the optimal one, that is, the electrolyte maintained in the reactor was too long, thereby overoxidation of formate to carbonate takes place. The catalytic results in one reactor at different flow rate was shown in Supplementary Fig. 20. The optimal formate selectivity and FE were obtained at 1.98 mL min^{-1} . To our expectation, when a lower flow rate was applied (1.89 and then 1.68 mL min^{-1}), higher glucose conversion but lower formate and FE were observed, in consistent with the above catalytic results when two tandem modules were used to replace one reactor.

2. As the reviewer suggested, using tandem modules would be efficient to improve the productivity and scalability of biomass electrooxidation technology. According to the above catalytic comparison, it is expected that similar catalytic results (conversion, product selectivity and FE) but with higher flow rate can be achieved in tandem modules compared with that in one reactor.

As proof-of-concept, we constructed a three-tandem-modules reaction system. The reaction was operated at a total current of 9 A (3 A for each module, revised Supplementary Fig. 46a). We obtained glucose conversion of 80.9% , formate yield of 66.0% and selectivity of 81.6% , at a flow rate of 6 mL min^{-1} (revised Supplementary Fig. 46b). In contrast, in a single reactor, we obtained similar catalytic performance (glucose conversion of 80.2% , formate yield of 67.2% and selectivity of 83.8% ; reactor system at condition of 3 A) but with much lower flow rate (1.98 mL min^{-1}). These results show high potential of linearly scaling up of tandem modules for biomass valorization.

Based on above results and discussion, we revised the Manuscript and Supplementary Information as follows:

“Furthermore, the reaction system can be linearly scaled up by tandem design without diminishing catalytic performances (see discussion in Supplementary Figs. 45, 46), showing great potential of assembling modules in parallel or in cascade for pilot-scale application (Supplementary Fig. 47)” (Please see Page 15 in the revised Manuscript)

“Under an optimized conditions (i.e., current of 3 A, flow rate of 1.98 mL min⁻¹, obtained from Supplementary Fig. 20), good catalytic results (80.2% of conversion, 83.8% of formate selectivity, and 89.6% of formate FE) were obtained after the electrolyte passed through the first module (Supplementary Fig. 45b). After the liquid stream passed through the second module, higher conversion efficiency (94%) was achieved, but formate selectivity (27.8%) and FE (16.8%) decreased. This can be explained by the overoxidation of formate to carbonate, as evidenced by the formation of large number of CO₂ when the electrolyte was acidified (pretreatment for HPLC analysis).” (Please see Page 39 in the revised Supplementary Information)

“The reaction was operated at a total current of 9 A (3A for each module, Supplementary Fig. 46a). We obtained glucose conversion of 80.9%, formate yield of 66.0% and selectivity of 81.6%, at a flow rate of 6 mL min⁻¹ (Supplementary Fig. 46b). In contrast, in a single reactor, we obtained similar catalytic performance (glucose conversion of 80.2%, formate yield of 67.2% and selectivity of 83.8%; reactor system at condition of 3 A) but with much lower flow rate (1.98 mL min⁻¹). These results show high potential of linearly scaling up of tandem modules for biomass valorization.” (Please see Page 40 in the revised Supplementary Information)

Revised Supplementary Figure 45. Catalytic performances of glucose electrooxidation in a two-tandem-module reaction system. a Configurations of reaction system consisting of two tandem modules. **b** Corresponding catalytic results after the electrolyte passed through the first and second module. Error bars correspond to the standard deviation of three measurements.

Supplementary Figure 20. Catalytic performances of the single-module SPCFR for GOR as function of flow rate at 3 A. Error bars correspond to the standard deviation of three measurements.

Revised Supplementary Figure 46. Catalytic performances of glucose electrooxidation in a three-tandem-module reaction system. a Scheme and configurations of the three-tandem-module reaction system. **b** Catalytic performances of the reaction system as a function of flow rate. Error bars correspond to the standard deviation of three measurements.

Comment 2: In Fig. 3b and line 253, what is the basis for absenting the ionic exchange membrane in the designed SPCFR system? Whether the reaction intermediate and target product of anodic oxidation will be reduced or hydrogenated at the cathode?

Response: This is a very important comment. We choose a membrane-free electrolyzer design based on the following considerations: (1) Glucose and its oxidation products might not be easily reduced under our reaction conditions (as demonstrated latter). (2) The organic substrates have negligible effects on hydrogen evolution reaction at cathode according to previous literatures (*J. Am. Chem. Soc.* 2016, 138, 13639–13646; *Nat. Commun.* 2020, 11, 265); (3) High-cost and stability issue of membranes, especially for anion exchange membrane.

To investigate if the reaction intermediates and/or products generated via anodic oxidation can be further reduced at cathode, we analyzed HPLC chromatograms of glucose-electrolysis product in an undivided cell and also in a H-cell (equipped with AEM). As reported in previous literature (*ACS Catal.* 2020, 10, 13895–13903), electrochemical reduction of sugars results in polyols at alkaline medium (pH>11). As shown in revised Supplementary Fig. 6c, HPLC analysis indicated that similar products distribution was obtained by using undivided cell and H-cell, without observing

associated reduction products (e.g., polyols) from glucose and intermediates in the undivided cell, suggesting that the reaction intermediates and/or products generated via anodic oxidation cannot be further reduced at cathode under our reaction conditions.

Inspired by the reviewer's comment, we consider that if the catalyst at cathode (nickel foam) is active enough for electroreduction, or if the organic substrates and/or products are inevitable to be reduced, an ionic exchange membrane should be present in the reactor to avoid electroreduction.

Based on the above discussion, we have revised the Supplementary Information as following:

“In addition, HPLC chromatograms show that similar products distribution was obtained by using undivided cell and H-cell, without observing associated reduction products (e.g., polyols) from glucose and intermediates (Supplementary Fig. 6c), suggesting that glucose and associated intermediates and products may not be reduced in the absence of membrane under our reaction conditions. Together with the negligible effects of organic substrates (i.e., HMF, glucose) on hydrogen evolution reaction at cathode according to previous literatures^{37,38} and high-cost and stability issue of membranes (especially for AEM), we adopted an undivided cell in the following studies.” (Please see Page 11 in the revised Supplementary Information)

Revised Supplementary Figure 6c HPLC chromatograms of catalytic products using an undivided cell and a H-cell at four-hours of reaction.

Comment 3: In Fig. 3d and line 271, is it possible for other by-products to be

formed since the FDCA selectivity is 92.3% not closes to 100%?

Response: We thank the reviewer for the insightful comment.

First, we apologize that the mentioned “92.3%” is carbon balance value, which was mistakenly wrote as FDCA selectivity in the original manuscript. The actual FDCA selectivity was measured to be 91.3%. The total selectivity of other detectable products is 0.70%, including DFF (0.13%), HMFCA (0.38%), and FFCA (0.19%), as shown in revised Supplementary Fig. 27.

Second, as the reviewer mentioned, other by-products were expected to generate according to the carbon balance results. We speculate that the carbon loss (that is 7.7%, according to carbon balance of 92.3%) is mainly attributed to the formation of humins, with fewer generation compared with the results in batch reactor, which is challenging to quantify.

Revised Supplementary Figure 27. Selectivity of detectable products and other unknown by-products. Inset enlarges the products with very low selectivity.

Based on the above discussion, we have revised the Manuscript as following:

“The results show high SPCE (96.6%) with high FDCA selectivity (91.3%) at a flow rate of 0.79 mL min⁻¹ and current of 3 A. The total selectivity of other detectable products is 0.70%, including DFF (0.13%), HMFCA (0.38%), and FFCA (0.19%). We speculate that the carbon loss (that is 7.7%, according to carbon balance of 92.3%) is mainly attributed to the formation of humins, with fewer generation compared with the results in batch reactor, which is challenging to quantify (Fig. 3e, Supplementary Figs.

26, 27).” (Please see Page 11 in the revised Manuscript)

Comment 4: In supplementary Figure 17, the conversion and formate selectivity increase as the flow rate decreases, which means that lower flow rate is beneficial to more formation of formate. However, the FE of formate shows the opposite trend at the same condition, why? The similar phenomenon also displays in supplementary Figure 21, why?

Response: We appreciate the reviewer for pointing the flow rate-dependent performance, which is very important. We response it from the following two aspects:

1. Regarding the increase of substrate (glucose and HMF) conversion and product (formate or FDCA) selectivity as flow rate decreases, it is attributed to the longer duration time of substrate and intermediates in the reactor at lower flow rate. In this context, more substrate and intermediates would be converted to corresponding final product (glucose to formate, HMF to FDCA), leading to higher product selectivity. This rationale was supported by the gradual decrease of intermediate arabinose with higher formate selectivity as the flow rate decreased during electrooxidation of glucose (the Supplementary Fig. 20). Similarly, during HMF electrooxidation to FDCA (the revised Supplementary Fig. 25b), the intermediate HMFCa and FFCA were gradually converted to FDCA as the flow rate decreased, leading to higher FDCA selectivity.
2. Regarding the lower Faradaic efficiency as flow rate decreases, it is attributed to the decrease of substrate and intermediates concentration in the electrolyte at low flow rate. As a result, oxygen evolution reaction become a competitive reaction over anode owing to insufficient mass transfer, resulting in the decrease of Faradaic efficiency during electrooxidation of glucose and HMF at lower flow rate. In addition, overoxidation of formate also contributed to the lower Faradaic efficiency during glucose oxidation. As shown in Supplementary Fig. 20, formate selectivity decreased when flow rate is lower than 1.98 mL min^{-1} , indicating that formate generation is diminished by the overoxidation.

Based on the above discussion, we have revised the Supplementary Information as

following:

“As the flow rate decreases, the duration time of substrate and intermediates in the reactor increases, leading to higher conversion of glucose and intermediates (such as arabinose). After optimizing flow rate, we obtained the highest formate yield (67.2%) and selectivity (83.8%) at 1.98 mL min⁻¹. At lower flow rate, the oxygen evolution and formate overoxidation become more competitive, resulting in the decrease of formate FE and selectivity.” (Please see Page 20 in the revised Supplementary Information)

Supplementary Figure 20. Catalytic performances of the single-module SPCFR for GOR as a function of flow rate at current of 3 A. Error bars correspond to the standard deviation of three measurements.

Supplementary Figure 25b Catalytic performances of the single-module SPCFR for HMF oxidation as a function of flow rate at current of 3 A.

Comment 5: It is suggested that the authors provide detailed characterizations of CoOOH/NF anode catalyst after stability testing of Fig. 3e.

Response: We thank the reviewer for this important comment. Based on this suggestion, the spent CoOOH/NF anode was characterized by SEM, XPS, and Raman techniques. As shown in revised Supplementary Fig. 31a, the structure of CoOOH/NF was maintained, showing similar nano-array structure to that of the fresh catalyst (Supplementary Fig. 4a). In addition, Co 2p XPS spectra and Raman spectra of the spent CoOOH were well consistent with that of the fresh one (revised Supplementary Fig. 31b, c), confirming the stability of CoOOH/NF anode.

Based on these results, we revised the Manuscript and Supplementary Information as follows:

“No obvious decay of catalytic performances and catalyst structure was observed for more than 100-hours at current of 5 A (Fig. 3f, Supplementary Figs. 30, 31).”
(Please see Page 11 in the revised Manuscript)

“After stability test (Fig. 3f), the spent CoOOH/NF anode was characterized by SEM, XPS, and Raman techniques. As shown in Supplementary Fig. 31a, the structure of CoOOH/NF was maintained, showing similar nano-array structure to that of the fresh catalyst (Supplementary Fig. 4a). In addition, Co 2p XPS spectra and Raman spectra of

the spent CoOOH were well consistent with that of the fresh one (Supplementary Fig. 31b, c), confirming the stability of CoOOH/NF anode.” (Please see Page 28 in the revised Supplementary Information)

Revised Supplementary Figure 31. Characterizations of spent CoOOH/NF anode. **a** SEM images of spent CoOOH/NF. Inset shows the enlarged region. **b** The Co 2p XPS spectra and **(c)** Raman spectra of fresh and spent CoOOH.

Comment 6: In supplementary Figure 31 and 32, the integration of NMR spectra is unclear, it is better to provide clearer one.

Response: We appreciate the reviewer for raising this valuable comment.

Supplementary Figures 31 and 32 have been replotted to improve the readability, as shown below:

Revised Supplementary Figure 37. ^1H NMR spectra of the electrolyte of $^{13}\text{C}1$ -labeled glucose at different charge of theoretical value. **a** 16.7%, **b** 33.3%, **c** 50.0%, **d** 83.3%.

Revised Supplementary Figure 38. ^1H NMR spectra of the electrolyte of $^{13}\text{C}1$ -labeled gluconate at different charge of theoretical value. **a** 16.7%, **b** 34.2%, **c** 49.2%, **d** 64.1%, **e** 82.0%, **f** 95.6%.

Comment 7: Some writing mistake can be found in Fig. 3b, such as “the mose adopted”. Please thoroughly check the full text carefully.

Response: We appreciate the meticulous comment. We have checked the manuscript

and supplementary information thoroughly, and corrected the mistakes.

Revised Fig. 4b Reported reaction pathways for C–C bond cleavage of polyhydroxy compounds.

Comment 8: What is the solubility of HMF like the high concentration up to 800 mM in the aqueous phase? Is a cosolvent needed?

Response: We thank the reviewer for this comment. As shown in the revised Supplementary Figure 57b, high-concentration HMF (800 mM or even up to 1.6 M) is miscible with water, giving a homogenous solution.

In addition, aqueous HMF solution with concentration of 15 wt.% (corresponding to ~1.19 M) was used as a feedstock solution in previous publication (*J. Appl. Electrochem.* 2018, 48, 611–626). Furthermore, 50 wt.% and 70 wt.% of aqueous HMF solutions are provided in commercial market (<http://www.guoshengtech.com/productinfo/909648.html>). Based on these results, we conclude that HMF has high solubility in water, and cosolvent is not needed.

Revised Supplementary Figure 57b Photographs of HMF aqueous solution at high concentration of 800 mM (left) and 1.6 M (right).

Comment 9: If the liquid stream flow rate ratio (i.e. the ratio of substrate to KOH) change via two peristaltic pumps with different flow rates in this SPCFR system, what will be happened for the performance? They were fixed to be same via one peristaltic pump like in supplementary Figure 36. Is this parameter studied in more detail?

Response: We thank the reviewer for providing a very insightful comment. As mentioned by the reviewer, varying the flow rate of substrate and KOH electrolyte by two peristaltic pumps is equal to change the ratio of substrate/KOH in the electrolyte that pass through the electrolyzer. To ease the operation, we consider a similar effect can be achieved by changing the concentration of substrate and KOH in the stock solutions with the same flow rate.

Therefore, we conducted glucose electrooxidation at a fixed glucose concentration (100 mM) but with different KOH concentrations (from 0.5 to 2 M) at the same flow rate controlled by a multichannel peristaltic pump. As shown in revised Supplementary Figure 21, inferior catalytic performances (e.g., FE and selectivity of formate are both <40%) were obtained in 0.5 M KOH electrolyte. This can be explained by the overoxidation of formate to carbonate, as large quantity of bubbles was generated when

the electrolyte was acidified for HPLC analysis. In contrast, good catalytic performances can be achieved (FE and selectivity of formate are both >80%) after KOH concentration was increased to 1 M. These results indicate that optimizing the ratio of glucose/KOH is important to obtain good catalytic performance. Theoretically, converting 0.1 M glucose gives 0.6 M formic acid, which consumes KOH and decreases the pH of electrolyte, hence high-concentration of KOH is needed to maintain the strong alkaline medium. Therefore, these results imply that strong alkali medium is conducive to glucose oxidation to format. More systematic investigations are required in the future study.

Based on these results, we revised the Manuscript and Supplementary Information as follows:

“To optimize the ratio of glucose/KOH, we evaluated the catalytic performances of glucose electrooxidation at a fixed concentration (100 mM) but with different concentration of KOH electrolyte (from 0.5 to 2 M). As shown in Supplementary Fig. 21, inferior catalytic performances (e.g., FE and selectivity of formate is <40%) were obtained in 0.5 M KOH electrolyte. This can be explained by the overoxidation of formate to carbonate, as large quantity of bubbles was generated when the electrolyte was acidified for HPLC analysis. In contrast, good catalytic performances can be achieved (the FE and selectivity of formate is >80%) after KOH concentration was increased to 1 M. These results indicate that optimizing the ratio of glucose/KOH is important to obtain high catalytic performance.” (Please see Page 22 in the revised Supplementary Information)

Revised Supplementary Figure 21. Glucose electrooxidation in different concentration of KOH electrolyte. **a** Catalytic performances of glucose electrooxidation. **b** Yield of various product. Reaction conditions: single-module SPCFR system, current: 3 A, flow rate: 1.98 mL min⁻¹, glucose concentration: 100 mM. The desired composition of electrolyte was controlled by one multichannel peristaltic pump to mix 200 mM glucose solution and KOH solution with different concentrations.

Comment 10: How does the gas-liquid separation chamber work in the stacked SPCFR system in supplementary Figure 36? What is the amount and purity of the hydrogen that generated and separated?

Response: We appreciate the reviewer for raising these insightful comments.

1. To clearly illustrate the working mechanism of the gas-liquid separation chamber, the scheme has been revised. As shown in revised Supplementary Fig. 43, there are two outlets in the gas-liquid separation chamber of the stacked SPCFR, with the upper one for gas stream and the lower one for liquid stream. In addition, there is a gate before the outlets. The gas (that is H₂) is flowing through the gate and exiting from the upper outlet because of its low density. The liquid (that is electrolyte) is flowing through the gate, falling down and exiting from the lower outlet. This design for separating gas and liquid is inspired by a previous work from Kato and colleagues (*Joule* 2021, 5, 687-705), which was originally designed for CO₂ electroreduction.

2. In the last version of the manuscript, the generated gas stream was directly pass through a gas analyzer without drying, giving H₂ purity of 97.69% (revised Supplementary Fig. 42b), which contains a small fraction of water vapor. To remove water vapor, we connected the gas outlet of the electrolyzer to a drying column with desiccant silica gel before entering the hydrogen analyzer (revised Supplementary Fig. 43). As a result, the H₂ purity was determined to >99.9%. In addition, H₂ productivity was calculated to be 279.8 mmol h⁻¹ when the reaction was operated at 15 A.

Based on above results and discussion, we revised the Manuscript and Supplementary Information as follows:

“As a result, formate solution (562.8 mM) with space-time-yield (STY) of 256.6 mmol h⁻¹ (corresponding to 11.8 g h⁻¹) coupling with H₂ production (>99.9% purity) with STY of 279.8 mmol h⁻¹ (corresponding to 0.56 g h⁻¹) were continuously produced using this stacked SPCFR system (Supplementary Figs. 43, 44), achieving co-production of biomass-derived valuable chemicals and H₂ fuel.” (Please see Page 15 in the revised Manuscript)

“There are two outlets in the gas-liquid separation chamber of the stacked SPCFR, with the upper one for gas stream and the lower one for liquid stream. In addition, there is a gate before the outlets. The gas (that is H₂) is flowing through the gate and exiting from the upper outlet because of its low density. The liquid (that is formate solution) is flowing through the gate, falling down and exiting from the lower outlet. This design for separating gas and liquid is inspired by a previous work by Kato and colleagues³⁷, in which a reactor was designed for CO₂ electroreduction. The generated gas from the electrolyzer passed through a drying column filling with desiccant silica gel before it entered to a H₂ analyzer. After drying treatment, H₂ purity was measured to be 99.99%, higher than that without drying (97.69%; Supplementary Fig. 42b).” (Please see Page 38 in the revised Supplementary Information)

Revised Supplementary Figure 43. Scheme of gas-liquid separation chamber in the stacked SPCFR electrolyzer. The generated gas from the electrolyzer passed through a drying column filling with desiccant silica gel before it entered to a H₂ analyzer.

Response to Reviewer #3:

I am very positive about the publication of this article. The data presented here is relevant to the field and the conclusions are well-supported by the experiments. Below are some comments:

Comment 1: During the oxidation of the biomass-derived molecules, green hydrogen is produced in the cathode. It called my attention that the authors have not mentioned the importance of the production of this energy vector.

Response: This is a very constructive comment. Based on this comment, the advantages of the paired electrolysis for coproduction of green hydrogen and valuable biomass-derived commodities have been discussed in the revised manuscript. Please see the reversion as below:

“Meanwhile, green hydrogen can be obtained at cathode via hydrogen evolution reaction, which is regarded as an important energy carrier of intermittent renewable energy^{26-29”} (Please see Page 3 in the revised Manuscript)

Comment 2: In this paper, the separation and quantification of the products are extremely relevant. The main drawback of this paper is that the authors did not clearly explain the HPLC protocol used to run the samples. They should also show which products are they able to indeed separate.

Response: We thank the reviewer for raising the important comment.

1. In this work, we adopted the HPLC analytical protocol developed by NREL to analyze a portfolio of biomass derivatives (<https://www.nrel.gov/docs/gen/fy08/42623.pdf>). Specifically, HPLC analysis was performed on Agilent 1260 II equipped with a H⁺ column (300 mm × 7.8 mm, IC Sep ICE-Coregel 87H3) at 60 °C using 5 mM H₂SO₄ as the mobile phase at a flow rate of 0.6 mL min⁻¹. The standard curves, suitable detector, and retention time for each compound have been summarized in the revised Supplementary Figs. 9, 10 and Supplementary Table 3 (shown below). In addition, humins generated via base-catalyzed HMF polymerization are difficult to be separated and quantified by HPLC

or other accessible techniques. Except humins, other substrates and products in this work can be separated by the H⁺ column and quantified by HPLC.

2. For practical separation, the dominant product in the generated electrolyte can be readily isolated. Formate is the dominant product in the electrolyte from polyhydroxy compounds (glucose, xylose, glycerol, etc.) electrooxidation, which can be possibly isolated by pressure-swing distillation after acidifying the electrolyte (*Nat. Catal.* 2022, 5, 268–276). In our work, the generated formate (in potassium salt form) was acidified by formic acid to produce potassium diformate (KDF) via decolorization, concentration, crystallization, filtration and drying processes (Supplementary Fig. 53). KDF is a value-added commodity as a feed additive for promoting animal growth. For the product of HMF electrooxidation, FDCA can be isolated by acidifying and filtration (Supplementary Fig. 30). Therefore, both formate and FDCA can be isolated by mature separation units.

Revised Supplementary Figure 9. Standard curves of polyhydroxy compounds and corresponding derivatives for HPLC analysis. Corresponding HPLC conditions are shown in Supplementary Table 3.

Revised Supplementary Figure 10. Standard curves of HMF and corresponding derivatives for HPLC analysis. Corresponding HPLC conditions are shown in Supplementary Table 3.

Revised Supplementary Table 3. HPLC conditions for quantification of biomass derivatives *a,b,c*

Compound	Detector	Retention time (min)
glucose	RI	9.5
sorbitol	RI	10.3
fructose	RI	10.2
gluconic acid	UV (210 nm)	9.2
xylose	RI	10.0
xylitol	RI	11.2
arabinose	RI	11.0
erythritol	RI	11.3
erythrose	RI	11.8
glycerol	RI	13.2
glyceric acid	UV (210 nm)	11.4

lactic acid	UV (210 nm)	12.7
glyceraldehyde	RI	11.6
tartronic acid	UV (210 nm)	8.6
ethylene glycol	RI	15.7
glycolic acid	UV (210 nm)	12.7
oxalic acid	UV (210 nm)	7.3
glycolic aldehyde	RI	12.6
formic acid	UV (210 nm)	14.1
formaldehyde	RI	13.7
HMF	UV (265 nm)	30.3
FDCA	UV (265 nm)	15.8
HMFCFA	UV (265 nm)	20.5
FFCA	UV (265 nm)	21.5
DFF	UV (265 nm)	37.8

^a According to previous protocol developed by NREL¹⁹⁵, HPLC was used for analyzing a portfolio of biomass derivatives.

^b The HPLC analysis was performed on Agilent 1260 II equipped with a H⁺ column (300 mm × 7.8 mm, ICSep ICE-Coregel 87H3) at 60 °C using 5 mM H₂SO₄ as the mobile phase at a flow rate of 0.6 mL min⁻¹.

^c Suitable detector and retention time for each compound are summarized.

Supplementary Figure 53. Procedures for potassium diformate (KDF) synthesis from the produced electrolyte from electrooxidation of pure glucose, lignocellulosic sugars, and crude glycerol. After electrooxidation, the generated formate (in potassium salt form) was acidified by formic acid to produce KDF via decolorization, concentration, crystallization, filtration and drying processes.

Comment 3: About the mechanism, you could mention something more than simply “providing experimental evidence for our previous theoretical calculation”. I think that the calculations you published are extremely valuable, but the level of detail of them is much deeper than what these results can show. Here you show that you break the C-C bond and the importance and reactivity of the aldehydes, which is really valuable, but you are not giving much evidence for mechanistic details. To link these observations (I mean, the products generated in different conditions) to the detailed mechanism is too much. What indeed can give stronger insights into the mechanism are, for example, Raman in situ experiments like that you performed in the same paper and/or a simpler experiment like that shown in figure 17 of this manuscript:

<https://chemrxiv.org/engage/chemrxiv/article-details/63f64e9032cd591f12534067>

The importance/reactivity of the aldehydes have been shown in several articles but

most of them use PGM electrodes. I am not suggesting including the references but maybe they help improving the discussion.

<https://www.sciencedirect.com/science/article/pii/S0013468618325581?via%3Dihub>

ub

Response: We thank the reviewer for the comment and also the valuable suggestions.

1. In our previous theoretical studies on glucose electrooxidation to formate (*Angew. Chem. Int. Ed.* 2023, e202219048), the calculation results indicate that the C–C bond cleavage is initiated at C1–C2 position to give arabinose and formate, owing to the smallest bond order among the five C–C bonds in glucose (as the figure shown below). To clearly demonstrate the specific coincidence between experimental findings in this work and previous theoretical results, we revised the manuscript as following:

“By combining above HPLC results and ¹³C1-labeling experiments (see discussion in Supplementary Note 3), we can exclude route II for glucose-to-formate, and rationalize that the reaction was initiated from C1–C2 bond position of glucose (Fig. 4a (up)). This is well agreement with our previous theoretical calculation that glucose electrooxidation is initiated at C1–C2 position to give arabinose and formate, owing to the smallest bond order among the five C–C bonds in glucose molecule⁴⁴ (see details in Supplementary Note 3).” (Please see Page 13 in the revised Manuscript)

Figure showing Mulliken bond order of the glucose and gluconic acid from previous publication (*Angew. Chem. Int. Ed.* 2023, e202219048)

2. We thank the reviewer for the valuable suggestion that giving more evidence to link the observations (i.e., aldehydes or acids) to the detailed mechanism. The suggested situ Raman experiments and electrochemical redox experiments from the recommended paper are useful tools. However, large numbers of aldehyde and acid intermediates with similar structure are involved in glucose-to-formate reaction, thus it is difficult for Raman spectroscopy to precisely discriminate the in-situ formed compounds. In addition, the suggested electrochemical redox experiment (<https://www.sciencedirect.com/science/article/pii/S0013468618325581?via%3Di> hub) is able to probe the reactivity of substrats, but unable to probe the specific reaction sites of one compound. To tackle this issue, we provided evidences from isotope labeling experiments and carbon balance analysis to link the observations (i.e., aldehydes or acids) to the detailed mechanism.

First, we performed isotope experiments using ^{13}C -labeled glucose (Supplementary Fig. 37) and gluconate (Supplementary Fig. 38) as the substrates, respectively. As shown in Supplementary Fig. 37, ^{13}C -labeled formate was

produced when ^{13}C -labeled glucose was used as the substrate, indicating that the aldehyde group in glucose was transformed into formate. In addition, the semiquantitative ^1H NMR analysis shows that the ratio of $^{13}\text{C}/^{12}\text{C}$ in the generated formate from ^{13}C -labeled glucose reached 0.3 (higher than the theoretical value 0.2) at low charge, and it successively decreased at longer reaction time (Supplementary Fig. 39). This trend can be explained by that the C1–C2 bond cleavage in glucose is more favourable, thus the ^{13}C -labeled aldehyde group is firstly converted into formate. This is in well agreement with our previous theoretical calculation that glucose electrooxidation is initiated at C1–C2 position to give arabinose and formate, owing to the smallest bond order among the five C–C bonds (*Angew. Chem. Int. Ed.* 2023, e202219048).

In contrast, the ratio of $^{13}\text{C}/^{12}\text{C}$ in the generated formate from ^{13}C -labeled gluconate is about 0.01 (equal to the natural abundance of $^{13}\text{C}/^{12}\text{C}$) in the whole electrolysis process (Supplementary Fig. 39). This result reveals that the acid group of gluconate cannot be converted to formate, thus route II can be excluded. These results further demonstrated that the aldehydes, rather than acids, are the real intermediates of C–C bond cleavage to formate in glucose electrooxidation.

Second, we analyzed the products during electrooxidation of glycolic acid, glyceric acid, lactic acid, and corresponding polyols, to identify if the carboxy group in aldonic acid can be converted to formate. As shown in Supplementary Fig. 40, excellent carbon balance (90–100%) were obtained when polyols were used as the substrate, much higher than the results when associated acids are used as the substrate (carbon balance of 50–63%). The most possibility is that during electrooxidation of the acid molecule, the carboxy group was converted to CO_2 (route III in Fig. 4b). Collectively, we can exclude glycolic acid and glyceric acid as the main intermediates in electrooxidation of glucose, glycerol, and other biomass derived polyhydroxy compounds.

Based on the above results, we have revised Supplementary Note 3 to clearly demonstrate the link between experimental observations and detailed mechanism as following:

“To validate our above hypothesis, we performed isotope experiments using ^{13}C -labeled glucose (Supplementary Fig. 37) and ^{13}C -labeled gluconate (Supplementary Fig. 38) as the substrates. As shown in Supplementary Fig. 37, ^{13}C -labeled formate was produced when ^{13}C -labeled glucose was used as the substrate, indicating that the aldehyde group in glucose was converted to formate. In addition, semiquantitative ^1H NMR analysis shows that the ratio of $^{13}\text{C}/^{12}\text{C}$ in the generated formate from ^{13}C -labeled glucose reached 0.3 (higher than the theoretical value of 0.2) at low charge, and it successively decreased at longer reaction time (Supplementary Fig. 39). This trend can be explained by that the C1–C2 bond cleavage in glucose is more favourable, thus the ^{13}C -labeled aldehyde group was firstly converted to formate. This is in well agreement with our previous theoretical calculation that glucose electrooxidation is initiated at C1–C2 position to give arabinose and formate, owing to the smallest bond order among the five C–C bonds⁹.”

“In contrast, the ratio of $^{13}\text{C}/^{12}\text{C}$ in the generated formate from ^{13}C -labeled gluconate is about 0.01 (equal to the natural abundance of $^{13}\text{C}/^{12}\text{C}$) in the whole electrolysis process (Supplementary Fig. 39). This result reveals that the acid group of gluconate cannot be converted to formate, thus route II can be excluded. Collectively, these results further demonstrated that the aldehydes, rather than acids, are the real intermediates of C–C bond cleavage to formate in glucose electrooxidation.” (Please see Page 4 in the revised Supplementary Information)

“Based on these findings, we proposed electrooxidation of glucose to formate follows a processive α -scission pathway via aldehyde as the key intermediates, rather than aldonic acids (red arrow flow in Fig. 4c).” (Please see Page 14 in the revised Manuscript)

Supplementary Figure 39. Ratio of $\text{H}^{13}\text{COOH}/\text{H}^{12}\text{COOH}$ in the electrolyte of $^{13}\text{C}1$ isotope experiments.

Supplementary Figure 40. Eliminating route II for C–C bond cleavage to formate.

a Conversion and product yield in electrooxidation of EG and glycolic acid. **b**

Conversion and product yield in electrooxidation of glycerol and glyceric acid. c
Conversion and product yield in electrooxidation of 1,2-propylene glycol and lactic acid. Reaction conditions: 100 mM substrate in 50 mL 1 M KOH electrolyte, CoOOH/NF and Pt foil as anode and cathode, respectively, at 1.5 V vs RHE.

Comment 4: Authors achieved the following “By preliminary optimization of current and flow rate, and by temperature management of feeding solution, we achieved high productivity of formate in electrooxidation of glucose, delivering high SPCE (81.8%), good formate selectivity (76.5%), and high FE (91.7%) at current of 15 A (Supplementary Table 4).” Considering 100% of faradaic efficiency, I think informing the green hydrogen productivity is also important.

Response: We agree with the reviewer. The productivity of hydrogen was calculated to be 0.56 g h⁻¹ at a current of 15 A with near 100% Faradaic efficiency. The manuscript has been revised to highlight the importance of co-production of green hydrogen:

“As a result, formate solution (562.8 mM) with space-time-yield (STY) of 256.6 mmol h⁻¹ (corresponding to 11.8 g h⁻¹) and H₂ (>99.9% purity) with STY of 279.8 mmol h⁻¹ (corresponding to 0.56 g h⁻¹) were continuously produced using this stacked SPCFR system (Supplementary Figs. 43, 44), achieving co-production of biomass-derived valuable chemicals and H₂ fuel.” (Please see Page 15 in the revised Manuscript)

Comment 5: In the large-scale electrosynthesis experiments, the authors state that the cell working temperature is around 70 °C and that they must work on temperature management. I think that instead of cooling down the system, it is maybe better to optimize the method at this relatively high temperature.

Response: This is a very constructive comment. Indeed, it is maybe better to operate the electrolysis (e.g., traditional water splitting) at relatively high temperature, especially for improving energy efficiency. However, a higher electrolyte temperature may accelerate the non-Faradaic degradation rate of unstable biomass platforms, resulting in lower selectivity to targeted electrocatalytic product.

To estimate the temperature effect on non-Faradaic degradation, we conducted

glucose degradation reaction in 1 M KOH at 70 °C without applying electrolysis. As shown in revised Supplementary Fig. 48a, more than 80% of glucose was consumed within 5 min, corresponding to a reaction rate of $2.68 \times 10^{-4} \text{ mol L}^{-1} \text{ s}^{-1}$. Finally, glucose was completely degraded within 30 min, mainly delivering lactate (46.3% yield) and humins as dark-brown pigments (revised Supplementary Fig. 48b, c). In contrast, at room temperature (25 °C), only 41.4% of glucose was consumed after 7-h reaction (Supplementary Fig. 12a), giving a much lower degradation rate ($1.64 \times 10^{-6} \text{ mol L}^{-1} \text{ s}^{-1}$). These results suggest that high temperature may dramatically increase non-Faradaic degradation rate by two orders of magnitude (specifically, 163 times shown in the revised Supplementary Fig. 48b), and result in low formate selectivity (<65%) for reactions without temperature management (entries 4–6 of Supplementary Table 5). Therefore, we recommended to manage the temperature of electrolyte to suppress degradation of unstable biomass derivatives during electrolysis, thereby improving the selectivity of targeted product via electrolytic transformation.

Furthermore, we agree with the reviewer that it is better to optimize the method at relatively high temperature for electrolyzing stable substrates, such as bio-ethanol, which is beneficial for lowering electricity consumption of H₂ production. (*Nat. Commun.* 2014, 5, 4036)

Revised Supplementary Figure 48. Influence of high temperature on glucose degradation. **a** Glucose (100 mM) degradation and products formation as function of reaction time at 70 °C in 1 M KOH solution. **b** Comparison of glucose degradation rate in 1 M KOH solution at room temperature (25 °C) and at 70 °C. **c** Photograph of the reaction solution after 30 min.

Based on above results and discussion, we revised the Manuscript and Supplementary Information as follows:

“Note that the electrolyte temperature increased under high current owing to Joule heating effect (e.g., ~70 °C at 15 A), thus non-Faradaic degradation was accelerated by two-orders of magnitude (specifically, 163 times; Supplementary Fig. 48), accounting for the low formate selectivity (<65%) for reactions without temperature management (entries 4–6 of Supplementary Table 5).” (Please see Page 15 in the revised Manuscript)

“To estimate the temperature effect on non-Faradaic degradation, we conducted glucose degradation reaction in 1 M KOH at 70 °C without applying electrolysis. As shown in Supplementary Fig. 48a, more than 80% of glucose was consumed within 5 min, corresponding to a reaction rate of $2.68 \times 10^{-4} \text{ mol L}^{-1} \text{ s}^{-1}$. Finally, glucose was completely degraded within 30 min, mainly delivering lactate (46.3% yield) and humins as dark-brown pigments (Supplementary Fig. 48b, c). In contrast, at room temperature (25 °C), only 41.4% of glucose was consumed after 7-h reaction (Supplementary Fig. 12a), giving a much lower degradation rate ($1.64 \times 10^{-6} \text{ mol L}^{-1} \text{ s}^{-1}$). These results suggest that high temperature may dramatically increase non-Faradaic degradation rate by two orders of magnitude (specifically, 163 times shown in the Supplementary Fig. 48b), and result in low formate selectivity (<65%) for reactions without temperature management (entries 4–6 of Supplementary Table 5). Therefore, we recommended to manage the temperature of electrolyte to suppress degradation of unstable biomass derivatives during electrolysis, thereby improving the selectivity of targeted product via electrolytic transformation.” (Please see Page 41 in the revised Supplementary Information)

Comment 6: Again, about the production of value-added molecules. The authors informed the concentration of products. As they know the flow in the cell, they could inform the mass of products obtained per unit of time, for instance, per hour.

Response: We appreciate the reviewer for this valuable suggestion, which makes the

productivity more understandable for a general reader. The mass of products obtained per unit of time was calculated and provided in the revised manuscript:

“As a result, formate solution (562.8 mM) with space-time-yield (STY) of 256.6 mmol h⁻¹ (corresponding to 11.8 g h⁻¹) coupling with H₂ production (>99.9% purity) with STY of 279.8 mmol h⁻¹ (corresponding to 0.56 g h⁻¹) were continuously produced using this stacked SPCFR system (Supplementary Figs. 43, 44), achieving co-production of biomass-derived valuable chemicals and H₂ fuel.” (Please see Page 15 in the revised Manuscript)

“To our delight, starting with 600 mM HMF, a record-high concentration (556.9 mM) of FDCA was produced with excellent selectivity (96.9%), presenting a STY of 76.2 mmol h⁻¹ (corresponding to FDCA productivity of 11.9 g h⁻¹).” (Please see Page 17 in the revised Manuscript)

Comment 7: To close, I would like to suggest the authors stop using Pt CE. The Pt atoms can migrate to the working electrode and deposit there.

Response: We agree with the reviewer’s comment on the Pt issue. We will stop using Pt as the counter electrode in the future work.

Reviewer #1 (Remarks to the Author):

The authors have satisfyingly addressed all the comments raised by all reviewers, and the quality of this work has been substantially improved. I recommend the publication of this work in its current form.

Reviewer #2 (Remarks to the Author):

The authors have addressed the reviewer's concerns using additional data and corresponding discussion in this manuscript. The quality of this revision is improved, and its publication is recommended.

Reviewer #3 (Remarks to the Author):

The authors have properly answered all the points raised by the reviewer. In my opinion, the article deserves to be published in a high-impact journal like this. The multiple insights of the article and the detailed experimental information will enormously contribute to the field.